# LASeR: Towards Diversified and Generalizable Robot Design with Large Language Models

**Junru Song[1], Yang Yang[2,\*], Huan Xiao[3], Wei Peng[2], Wen Yao[2,\*], Feifei Wang[3,4]**
[1]School of Computer Science, Shanghai Jiao Tong University
[2]Intelligent Game and Decision Laboratory
[3]School of Statistics, Renmin University of China
[4]Center for Applied Statistics, Renmin University of China
`songjunru@sjtu.edu.cn, bigyangy@gmail.com`
`{xiaohuan001, feifei.wang}@ruc.edu.cn`
`{weipeng0098, wendy0782}@126.com`

## Abstract

Recent advances in Large Language Models (LLMs) have stimulated a significant paradigm shift in evolutionary optimization, where hand-crafted search heuristics are gradually replaced with LLMs serving as intelligent search operators. However, these studies still bear some notable limitations, including a challenge to balance exploitation with exploration, often leading to inferior solution diversity, as well as poor generalizability of problem solving across different task settings. These unsolved issues render the prowess of LLMs in robot design automation largely untapped. In this work, we present LASeR – Large Language Model-Aided Evolutionary Search for Robot Design Automation. Leveraging a novel reflection mechanism termed DiRect, we elicit more knowledgeable exploratory behaviors from LLMs based on past search trajectories, reshaping the exploration-exploitation tradeoff with dual improvements in optimization efficiency and solution diversity. Additionally, with evolution fully grounded in task-related background information, we unprecedentedly uncover the inter-task reasoning capabilities of LLMs, facilitating generalizable design processes that effectively inspire zero-shot robot proposals for new applications. Our simulated experiments on voxel-based soft robots showcase distinct advantages of LASeR over competitive baselines. Code at `https://github.com/WoodySJR/LASeR`.

## 1 Introduction

In recent years, Large Language Models (LLMs) have demonstrated remarkable reasoning, decision making, and generalization capabilities (Achiam et al., 2023; Touvron et al., 2023; Team et al., 2023; Team, 2023), sparking a flurry of research interest in their application to optimization problems. Earlier efforts embarked on leveraging LLMs to aid traditional search heuristics within evolutionary algorithms (EAs), such as selecting parent solutions for mutation and crossover (Liu et al., 2024a; Ye et al., 2024) or serving as surrogate models and candidate samplers in Bayesian Optimization (Liu et al., 2024b). More and more recent studies have explored the use of LLMs as "intelligent search operators". By receiving previously found solutions through prompts, LLMs effectively draw upon their in-context learning and pattern completing abilities to iteratively propose improved candidate solutions (Brahmachary et al., 2024; Huang et al., 2024b; Yang et al., 2024; Morris et al., 2024; Romera-Paredes et al., 2024; Lange et al., 2024). Such LLM-aided evolutionary frameworks have shown great promise in minimizing reliance on handcrafted search heuristics, facilitating convenient problem specification in natural language and rendering evolutionary processes more interpretable. To date, they have showcased proficiency in classic optimization problems such as the Traveling Salesman Problem and numerical functions, (Liu et al., 2024a; Brahmachary et al., 2024; Huang

---

[\*]Corresponding authors.

et al., 2024a), as well as real-world scenarios spanning code generation (Morris et al., 2024; Romera-Paredes et al., 2024), robotic control (Lange et al., 2024), protein design (Tran & Hy, 2024), etc.

Despite the promising results, we contend that existing studies exhibit two major limitations. For one, as many of them have noted that LLMs often struggle to balance exploration and exploitation and yield inferior solution diversity (Huang et al., 2024b; Tran & Hy, 2024), only expedient measures have been taken to address this issue, including adjustments to the temperature parameter (Yang et al., 2024; Liu et al., 2024a; Pluhacek et al., 2024; Ma et al., 2024) or utilizing pre-existing natural selection techniques such as binary tournament selection and "island models" (Qiu et al., 2024; Romera-Paredes et al., 2024). It remains to be investigated whether the reasoning capabilities of LLMs could be further harnessed to guide more intelligent exploratory behaviors in the search space. For the other, current LLM-aided evolutionary approaches generally lack a strong connection to the specific nature of real-world problems, which leads to suboptimal performances and solutions that can not generalize well.

Recently, LLMs have also made their way into the realm of robot design automation. Robot design automation represents a persistent challenge in modern robotics that aims to evolve robot morphology with minimal human intervention (Hu et al., 2022; 2023; Song et al., 2024a). However, related work is sparse and only represent rudimentary attempts. To our best knowledge, the only pertinent studies are Zhang (2024), Qiu et al. (2024) and Lehman et al. (2023). While Zhang (2024) utilizes LLMs to tune the hyperparameters of traditional EAs, the latter two pioneer the use of LLMs as search operators for robot design. Nonetheless, they bear the same limitations as listed above, which greatly hinder the application of LLMs to robot design automation. In particular, with growing interest in soft robots due to their versatility and biomimetic properties, their vast design spaces and intricacy of interaction dynamics among body parts cause existing search algorithms to generally fall short. This highlights the need for more judicious exploration that navigates a variety of design options while ensuring progressive enhancement in functionality (Bhatia et al., 2021; Shah et al., 2021; Song et al., 2024a; Saito & Oka, 2024). Furthermore, as it is common to have access to a repository of pre-designed robots from related tasks when designing for new applications, it is highly relevant to explore the inter-task reasoning capabilities of LLMs to facilitate positive transfer of prior design experience, thus fostering more generalizable design processes.

To address the aforementioned limitations, here we propose **LASeR** – Large Language Model-Aided Evolutionary Search for Robot Design Automation. LASeR distinguishes itself from previous LLM-aided evolutionary frameworks with a more delicate exploration strategy and generalizable optimization processes. Specifically, we present a novel *Diversity Reflection Mechanism* termed **DiRect**, which strategically instructs an LLM to reflect upon previously generated designs and suggest viable modifications to enhance diversity while preserving essential functional substructures. This mechanism thus fosters more knowledgeable exploratory behaviors that closely align with task objectives. Furthermore, by exploiting the abundant descriptive information available in robotic tasks, we not only yield substantially accelerated convergence to high-performing designs, but also unprecedentedly uncover the potential of LLMs to reason across different tasks and assimilate prior design experience for zero-shot robot proposals in new tasks.

To summarize, our contributions are as follows: (i) By interleaving evolutionary processes with diversity-oriented reflective thinking, we reshape the exploration-exploitation tradeoff of LLM-aided evolution with simultaneous improvements in solution diversity and optimization effciency. The former is particularly relevant for enhancing the robustness of robotic systems in volatile environments. (ii) With evolution firmly grounded in the background information of optimization tasks, we unlock the inter-task reasoning capabilities of LLMs in evolutionary computation, hopefully inspiring future work to further promote the generalizability of LLM-aided evolution across different problem settings. (iii) By unleashing the prowess of LLMs for robot design automation, we also aim to inspire future work that synergizes both robotic design and control with LLMs, achieving closed-loop development of embodied agents.

## 2 RELATED WORK

**Large Language Models as Evolutionary Search Operators.** Large Language Models (LLMs) represent a class of deep generative neural networks comprising billions or trillions of parameters and pretrained on web-scale texual data. In recent years, LLMs have demonstrated impressive rea-

soning, decision making, and generalization capabilities (Achiam et al., 2023; Touvron et al., 2023; Team et al., 2023; Team, 2023), which have sparked a flurry of research into exploiting them for optimization problems (Huang et al., 2024b; Wu et al., 2024). By receiving history search trajectories from the prompt (or context), LLMs have demonstrated effectiveness as pattern completion engines (Mirchandani et al., 2023), proposing improved solutions and facilitating evolutionary optimization through iterative interactions. Moreover, LLMs are adept at conditioning problem-solving processes on various kinds of prior knowledge expressed in natural language, without needing tedious mathematical formulations (Song et al., 2024b). All these favorable attributes position LLMs as promising substitutes for the manually designed search heuristics in traditional evolutionary algorithms (EAs), acting as novel, *intelligent* search operators. Since Lehman et al. (2023) introduced this LLM-aided evolutionary paradigm, subsequent studies have extended its methodology and showcased its proficiency in classic optimization tasks like the Traveling Salesman Problem (TSP) and numerical functions (Liu et al., 2024a; Brahmachary et al., 2024; Huang et al., 2024a), as well as practical problems spanning prompt optimization (Guo et al., 2023; Yang et al., 2024), code generation (Meyerson et al., 2023; Morris et al., 2024; Romera-Paredes et al., 2024), robotic control (Lange et al., 2024), etc. However, we argue that the use of LLMs as search operators is still in its early stage, with much of their potential untapped. Notably, existing studies have focused solely on single-task optimization, overlooking the intriguing possibility of LLMs to transfer experience across different tasks. Additionally, although LLMs have been shown to trail behind traditional EAs in balancing exploration and exploitation (Huang et al., 2024a; Tran & Hy, 2024), this nuanced aspect has received limited attention from previous research. We aim to tackle these limitations in this work.

**Robot Design Automation.** As Artificial Intelligence (AI) continues to revolutionize academia and industry, there is an increasing focus on integrating the perceptual and planning capabilities of multi-modal foundation models into various physical embodiments capable of interacting with their environments – a research field known as *Embodied AI* (Roy et al., 2021; Liu et al., 2024c). These advancements highlight the significance of autonomous robot design. Earlier works on robot design automation relied on traditional evolutionary algorithms and primarily targeted rigid robots (Sims, 1994; Chocron & Bidaud, 1997; Leger, 2012; Wang et al., 2019). In recent years, modular soft robots have garnered broad attention due to their flexibility, expressiveness, and biomimetic characteristics (Hiller & Lipson, 2011; Bhatia et al., 2021; Medvet et al., 2021). However, these advantages are accompanied by a combinatorially vast design space, necessitating more efficient search algorithms (Cheney et al., 2014). Consequently, an emerging line of research resorts to the estimation-of-distribution algorithms (EDAs) to enhance sample efficiency by explicitly tracking the distribution of high-performing robot designs. These approaches further leverage deep generative models, such as Generative Adversarial Networks (GANs; Goodfellow et al., 2020) and Variational Autoencoders (VAEs; Kingma, 2013), to bolster the representational capacity of EDAs (Hu et al., 2022; Song et al., 2024a). Despite their promising results, these models still require problem-specific mathematical formulation and neural architecture design, which is highly dependent on domain expertise and poorly generalizable. In this respect, Large Language Models, with their strong in-context learning abilities and extensive prior knowledge, hold the promise to transform the robotic design process (Stella et al., 2023). Nevertheless, the exploration of LLMs in this respect is sparse and warrants further investigation (Lehman et al., 2023; Zhang, 2024; Qiu et al., 2024).

## 3 LASeR: LLM-Aided Evolutionary Search for Robot Design Automation

In this section, we first present an overview of our algorithm, and then delve into the details of our prompt design and the novel *Diversity Reflection Mechanism*. Subsequently, we describe how LLMs can be instructed to facilitate effective knowledge transfer across different tasks, followed by a brief introduction to our fitness evaluation protocols.

### 3.1 Algorithm Framework

As illustrated in Figure 1(a) and detailed by Algorithm 1 in Appendix P, we integrate an LLM into the bi-level optimization framework commonly employed in robot design automation. Specifically, the inner loop optimizes a controller for each robot morphology through reinforcement learning, with the resulting task performance serving as the fitness evaluation. The outer loop evolves a

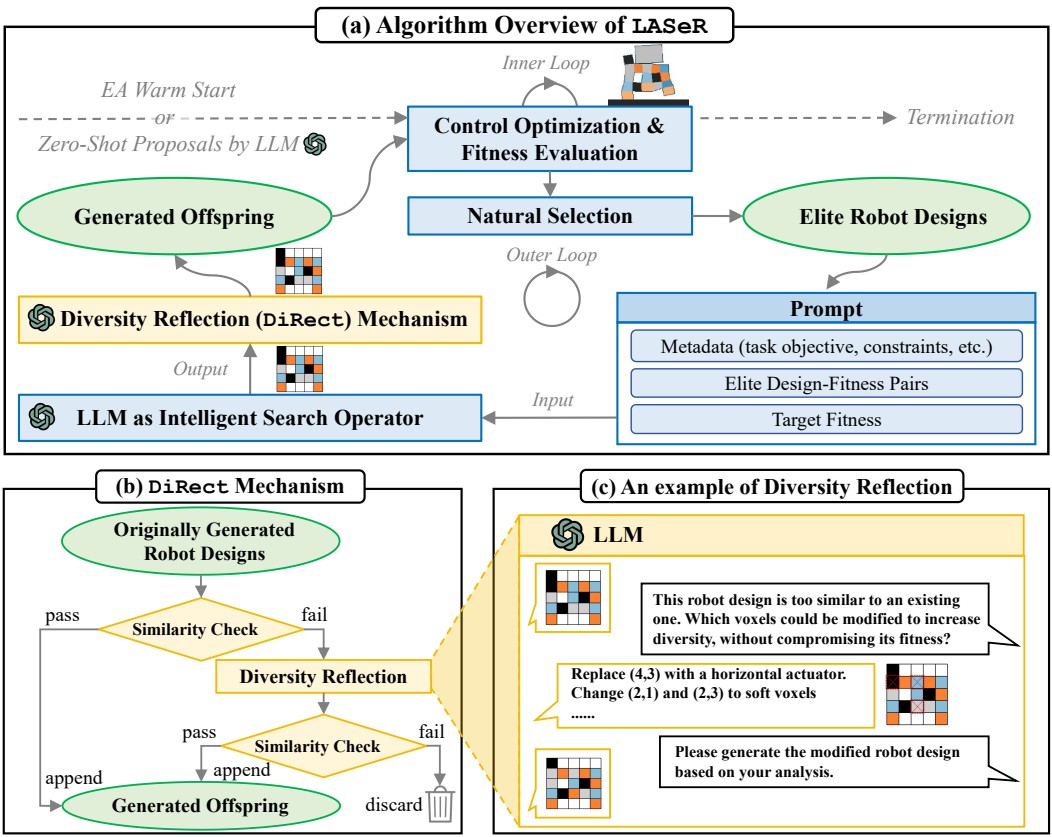

Figure 1: (a) algorithm overview of LASeR; (b) the Diversity Reflection (DiRect) Mechanism; (c) an example illustrating how Diversity Reflection works on a carrying robot. The illustration takes voxel-based soft robots (VSRs; Bhatia et al., 2021) as an example. Each VSR is represented as a two-dimensional material matrix, where each entry, ranging from 0 to 4, represents the material type at the corresponding position. ■, ▨, ▧, ▨ and □ denote rigid voxels, soft voxels, horizontal actuators, vertical actuators and empty voxels, respectively.

population of robot morphologies by carrying out natural selection and generating new offspring solutions in each generation. Here, instead of traditional evolutionary algorithms (EAs) that rely on manually designed search heuristics for generating offspring, an LLM is properly prompted to be our search operator. This is achieved by providing the LLM with previously evaluated robots and various kinds of metadata as context. However, we still bootstrap the evolutionary process with a few generations of conventional EAs before letting the LLM play its part, as this warm start would provide the LLM with an initial momentum (*i.e.,* improving directions) to build upon. Evolution terminates when a maximum number of robot evaluations is reached. We term our approach LASeR, short for LLM-Aided Evolutionary Search for Robot Design Automation. The following subsections elaborate on the key components of LASeR. Note that while the introduction takes voxel-based soft robots (VSRs) proposed in Bhatia et al. (2021) as an example, our approach is readily applicable to other robot types. For a detailed introduction to VSRs, please refer to Bhatia et al. (2021).

## 3.2 PROMPT DESIGN

As discussed in Section 3.1, the LLM is prompted to propose new offspring designs in each generation. As effective prompt design is crucial to elicit desired responses from LLMs, we craft a well-structured prompt comprising the following essential components:

- **Task-related metadata:** this part includes various auxiliary information to in-context adapt the LLM to serve as a search operator, including task objectives, descriptions of

simulation environment, constraints on robot designs, etc. This component is largely derived from the official documents of EvoGym (Bhatia et al., 2021). This metadata is largely missing in previous studies. We would showcase in our experiments that by fully grounding the evolutionary process on task-related metadata, we not only achieve more efficient design optimization, but also elicit inter-task reasoning capabilities from LLMs that greatly boost the generalization of design experience.

- **Elite design-fitness pairs:** this part includes robot designs that have survived natural selection, each accompanied by its fitness score. These designs are sorted in ascending order of fitness, so as to make it easier for the LLM to infer continuation patterns and extrapolate to potentially improved designs.

- **Target fitness:** we instruct the LLM to extrapolate from the sequence of elite designs and generate new designs that meet a pre-specified fitness score (e.g. 120% of current maximal fitness). This is referred to as the "just-ask" query in previous studies (Lim et al., 2024).

Additionally, we include some basic requirements into the system prompt to better align the behavior of the LLM with our intention. For instance, the LLM is strictly demanded to output new robot designs in numpy array format, enclosed between <solution> and </solution>, to allow for easier parsing. Even so, LLMs are still not guaranteed to generate valid robot designs in every interaction. To keep runtime and cost under control, we set an upper limit on interactions; once the limit is reached, we revert to a conventional EA to generate the remaining offspring. For full prompts please refer to Appendix A.

## 3.3 DiRect: Diversity Reflection Mechanism

The exploration-exploitation tradeoff is a longstanding dilemma in evolutionary computation. Properly timed exploratory behavior would contribute to search processes that not only are less prone to getting stuck in local optima, but also produce more diversified solutions. The latter property is particularly relevant for developing robotic systems in dynamic environments, where first-choice robots, once fail, must be immediately replaced with alternatives. However, it has been shown that LLMs often struggle to balance exploration and exploitation, yielding inferior solution diversity to existing EAs (Huang et al., 2024a; Tran & Hy, 2024). While many previous studies addressed this issue by tuning the temperature parameter of LLMs or resorting to diversity-preserving selection techniques, in this work we introduce a novel *Diversity Reflection Mechanism* (DiRect) that leverages the reasoning capabilities of LLMs to guide exploratory behaviors.

The idea of DiRect is straightforward. As depicted in Figure 1(b), for each newly proposed robot design, we assess its similarity to previously evaluated designs with a probability $p$. The new design is said to fail the similarity check if it shares more than $s$ voxels with at least one existing robot. In this case, the DiRect mechanism is triggered. Specifically, the LLM is first prompted to suggest modifications to voxels that could enhance variability without compromising fitness, with existing high-performing examples as reference. The LLM is then asked to return the modified robot design according to these suggestions. In section 4 we show that this reflection mechanism fosters more beneficial exploratory behavior in the search space, leading to more diversified robot designs while maintaining relevance to the task objective. Figure 1(c) displays a specific example where DiRect helps to modify a newly proposed carrying robot. The similarity threshold $s$ is an important hyperparameter that controls the performance of LASeR. We include general principles for choosing $s$, supported by experimental evaluations, in Appendix L.

## 3.4 LLM for Inter-Task Knowledge Transfer

The algorithm described thus far treats the robot design of each task independently, starting with a randomly initialized population. However, it is often the case that we already have access to a repository of pre-designed robots from existing tasks when designing for a new one. Under this circumstance, leveraging the prior design experience in one way or another would hopefully afford a boost in sample efficiency. However, to discern functional substructures from highly abstract robot morphologies (e.g. voxel-based soft robots), as well as to speculate which substructures will benefit a new task, poses a major challenge to humans. Our work represents a pioneering effort to exploit the reasoning capabilities of LLMs for this purpose. Specifically, by specifying the characteristics of task A and B, along with a collection of high-performing robot designs from task A, we instruct an

LLM to analyze the similarities and differences between the two tasks and infer potentially favorable substructures (such as specific patterns of voxel assembly) for task B. Based on this analysis, the LLM then proposes robot designs for task B, thus enabling *zero-shot* robot proposals (where "zero-shot" means that no evaluated robot samples from the new task are required). These robot proposals can then serve as an *informative* initialization to initiate further design search for task B.

## 3.5 FITNESS EVALUATION

In this work, we use the Proximal Policy Optimization (PPO) algorithm to optimize a separate controller for each robot design. The PPO algorithm enhances conventional gradient-based reinforcement learning algorithms by incorporating importance sampling into gradient estimation, allowing for the reuse of sample trajectories across multiple parameter updates. The PPO algorithm alternates between two key phases – data collection and policy update – until a predefined number of iterations are completed. With an optimized controller that maps environmental observations to appropriate actuation signals, we measure the fitness of a robot by calculating the cumulative reward it receives over a complete episode, which reflects its performance in accomplishing a given task. For further details on the PPO algorithm, please refer to Schulman et al. (2017).

## 4 EXPERIMENTS

We begin this section with an introduction to our experimental setups, and then analyze the results of our comparison and ablation studies in detail. Our experiments are designed to address the following questions:

- **Q1:** Can LASeR outperform state-of-the-art baselines in robot design automation?

- **Q2:** To what extent does DiRect improve the exploration-exploitation tradeoff of LLM-aided evolution?

- **Q3:** Does task metadata bring additional benefits to single-task robot design automation? Moreover, does it aid inter-task experience transfer and enable zero-shot robot design for new tasks?

- **Q4:** Previous studies have shown that different temperature parameters and versions of LLMs yield varying evolutionary outcomes. What are the specific impacts of these factors in our context?

## 4.1 EXPERIMENTAL SETUPS

**Benchmark Setting.** We base our experiments on Evolution Gym (EvoGym; Bhatia et al., 2021), a simulation environment designed for voxel-based soft robots (VSRs). In EvoGym, VSRs are represented in a grid-like layout and consist of five types of voxels: rigid voxels, soft voxels, horizontal actuators, vertical actuators, and empty voxels. VSRs achieve motion control by altering the volumes of actuators either horizontally or vertically according to action signals. For benchmarking of single-task optimization, we select four task instances: one locomotion task, Walker-v0, and three manipulation tasks, Carrier-v0, Pusher-v0 and Catcher-v0. For experiments of inter-task knowledge transfer, we use BridgeWalker-v0 and UpStepper-v0. A detailed introduction to these tasks is provided in Appendix B. For more information on EvoGym, please refer to Bhatia et al. (2021).

**Baselines.** We compare our method against the following baselines: **(i) Bayesian Optimization (BO)** (Kushner, 1964; Mockus, 1974)), a classic algorithm for optimizing expensive-to-evaluate functions. It employs a probabilistic model as the surrogate function and samples candidate solutions based on predicted mean and uncertainty. **(ii) Speciated Evolver (SE)** (Medvet et al., 2021)), a variant of the genetic algorithm (GA; Michalewicz, 2013) that divides the population into species to preserve diversity and prevent premature convergence. **(iii) RoboGAN** (Hu et al., 2022), an estimation-of-distribution algorithm (EDA) that utilizes the Generative Adversarial Network (GAN) to track the distribution of high-performing robot designs and generate new candidate solutions. **(iv)** The last baseline, which we term **LLM-Tuner**, is adapted from Zhang (2024) that uses LLMs to supervise the hyperparameter tuning of a genetic algorithm. Drawing comparison with LLM-Tuner would directly verify the benefits of LLMs serving as intelligent search operators. We additionally draw comparisons with two latest generative model-based evolutionary algorithms, MorphVAE (Song et al., 2024a) and OPRO (Yang et al., 2024), with results presented in Appendix F.

**Evaluation Metrics.** We employ the following metrics to evaluate the performance of various approaches: **(i) Maximal Fitness**, defined as the fitness of the best-performing robot design achieved within a specific number of evaluations. This metric is commonly used in robot design automation to assess optimization efficiency. **(ii) Diversity**: Given the significance of developing diverse robotic ecosystems to handle volatile environments, we measure the diversity of high-performing robot designs[1] from two perspectives: one is the average edit distance among all pairs of high-performing robot designs (Saito & Oka, 2024), and the other is the total number of distinct high-performing robot designs. We further aggregate the two values via weighted averaging, where the latter is multiplied by 0.1 so that they are roughly on the same scale and given equal importance. Please refer to Appendix J for a detailed discussion on diversity measurement. We also include an analysis of computational efficiency in Appendix O.

**Implementation Details.** We use GPT-4o-mini for both LASeR and LLM-Tuner, with the temperature parameter set as 0.7. For ablation studies, we additionally try out GPT-3.5-Turbo and temperatures of 1 and 1.5. Following the common practice in previous VSR studies (Song et al., 2024a; Saito & Oka, 2024; Dong et al., 2023; Bhatia et al., 2021), we choose the simple yet effective control protocol for fitness evaluation, *i.e.,* Multilayer Perceptron (MLP) as the controller for each robot design and PPO algorithm (Schulman et al., 2017) for policy training. Following previous studies on VSR design (Song et al., 2024a; Saito & Oka, 2024; Dong et al., 2023; Bhatia et al., 2021), robot designs are constrained to a $5 \times 5$ bounding box for an expressive yet tractable search space. Nevertheless, as demonstrated in Appendix G, our approach is scalable to larger design spaces. For fair comparison, each method is permitted 1000 robot evaluations. Experimental results of comparative studies are averaged across five independent runs. Our experiments are conducted on a server equipped with Intel Xeon processors running at 2.20 GHz and four NVIDIA Tesla RTX GPUs, with the system operating under Ubuntu 22.04. We relegate additional parameter settings to Appendix C. For further implementation details, please refer to our code repository[2].

## 4.2 COMPARISON STUDIES

### 4.2.1 SINGLE-TASK OPTIMIZATION

We begin our analysis by examining single-task optimization performances. As demonstrated in Figure 2, LASeR nearly consistently outperforms all baselines across the three tasks with significant margins. Specifically, LASeR achieves rapid convergence speeds to optimal robot designs, with only one exception on Walker-v0, where LLM-Tuner demonstrates slightly faster convergence in the early stage of evolution but ends up further from optimality. The superior performance of LASeR compared to LLM-Tuner highlights that LLMs have more important roles to play beyond merely tuning hyperparameters for traditional EAs. Results of significance tests are in Appendix D.

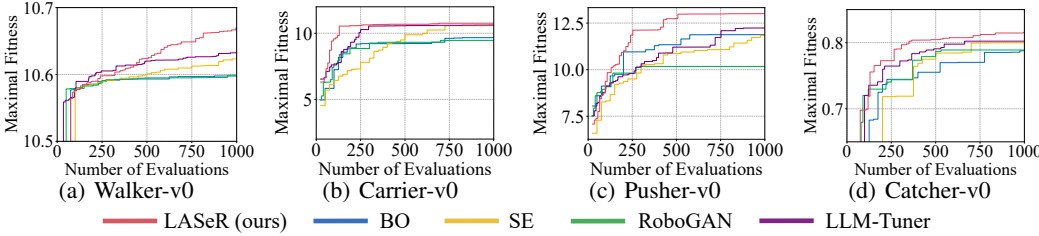

Figure 2: Comparative results of single-task optimization efficiency.

Table 4.2.1 further demonstrates the diversity of high-performing robot designs achieved by different methods. We observe that LASeR surpasses all baselines in three of the four tasks. We additionally compare the fitness performance of robot designs before and after being modified by DiRect, and find no significant difference (see Appendix E). This suggests that our Diversity Reflection Mechanism indeed encourages the LLM to introduce variability into robot design while keeping

---

[1]We first calculate the 90% quantile of fitnesses obtained by all methods, and consider robot designs with fitness exceeding this threshold as high-performing.

[2]https://github.com/WoodySJR/LASeR

its functionality largely intact. These results validate the distinct advantage of DiRect to promote beneficial exploratory behaviors directed towards high-performing regions. It is worth noting that the Bayesian Optimization algorithm, known for a balance between exploration and exploitation in its acquisition function, actually compromises a great deal of optimization efficiency for exploration and fails to generate high-performing robots in many cases. On the contrary, LASeR reshapes the exploration-exploitation tradeoff of LLM-aided evolution to yield dual benefits in optimization efficiency and diversity. For separate results of the edit distance and the number of high-performing designs, please refer to Appendix J. However, we note that the diversity of LASeR in Catcher-v0 is not as competitive. We suspect this is due to task complexity. Specifically, for more challenging tasks like Catcher-v0, while LLMs are still capable of extrapolating from existing solutions (as shown in Figure 2(d)), they struggle to recognize finer-grained functional structures. As a result, they might have difficulty introducing variability without compromising performance. We believe that helping LLMs better understand the roles of different parts within robot morphology—such as by incorporating images/videos of robots interacting with the environment—could be a promising direction for future research.

Table 1: Comparative results of diversity (reported as mean (std)).

|  | Walker-v0 | Carrier-v0 | Pusher-v0 | Catcher-v0 | Average Rank |
|---|---|---|---|---|---|
| BO | N/A | 11.88 (2.77) | 13.68 (2.10) | **19.76 (1.10)** | 3 |
| SE | 5.24 (0.26) | 15.26 (1.52) | 7.26 (3.91) | 13.91 (2.27) | 3.25 |
| RoboGAN | N/A | 10.94 (N/A) | N/A | 18.27 (1.44) | 4 |
| LLM-Tuner | 11.60 (4.35) | 18.26 (6.00) | 14.17 (6.83) | 13.89 (7.32) | 2.5 |
| **LASeR (ours)** | **23.09 (5.33)** | **20.87 (4.27)** | **20.91 (8.85)** | 8.07 (2.48) | **2** |

Note: When no more than one high-performing robot design is produced, diversity cannot be calculated. When this is the case across all repeated trials (e.g. BO on Walker-v0), the result is reported as "N/A". When high-performing robots emerge in only one trial (e.g. RoboGAN in Carrier-v0), the standard deviation is unavailable and reported as "N/A".

### 4.2.2 INTER-TASK KNOWLEDGE TRANSFER

Now we proceed to examine the ability of LLMs to transfer design experience across different tasks. To achieve this purpose, we introduce two more tasks: BridgeWalker-v0 and UpStepper-v0. Specifically, both BridgeWalker-v0 and UpStepper-v0 bear some resemblance to Walker-v0, but differ in their terrains: BridgeWalker-v0 involves locomotion on a soft rope-bridge, whereas UpStepper-v0 requires climbing stairs of varying lengths. The LLM is prompted to generate robot designs for each new task, given elite Walker-v0 robot designs. As shown in Figure 3(b)[3], the zero-shot proposals by LLM outperform both randomly generated designs and elite Walker-v0 designs evolved by LASeR, in terms of accomplishing the new tasks. This serves as sound evidence that the LLM is not simply replicating examplars in its context, but rather assimilating design experience that is beneficial for new settings. This is largely owing to our incorporation of task-related metadata that provokes inter-task reasoning within the LLM. For illustration, Figure 3(a) demonstrates some insights that the LLM drew from Walker-v0 elites to transfer to BridgeWalker-v0.

The zero-shot proposals are then leveraged as the initial population for further optimization. Figure 3(c-1) and 3(c-2) demonstrate that this informative initialization results in faster evolution than starting from scratch, and pulls away from baseline algorithms with even greater advantage. Also note that the zero-shot proposals for BridgeWalker-v0 turn out to be already near optimal before undergoing marginal improvement with evolution. These promising results unprecedentedly uncover the possibility of generalizable evolutionary processes driven by LLMs and hopefully inspire closer investigation in future work. Please note that while here we focus on intuitively similar task instances, we show in Appendix M that this is not strictly necessary for successful experience transfer.

---

[3]The result is averaged over ten robot designs in each case.

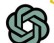

To design high-performing robots for the "BridgeWalker" task, we can leverage insights from the "Walker" designs. The primary differences between the two tasks are that "BridgeWalker" requires navigating on a soft rope bridge, which will behave differently under load due to its compliance and deformation characteristics. This suggests that the robot's structure should be optimized to distribute weight effectively and maintain stability while moving.

Here are some key insights drawn from "Walker" designs:
**1. Use of Soft Voxels:** These allow for elastic deformation, which is crucial on a soft surface like a rope bridge.
**2. Actuators Placement:** Horizontal and vertical actuators should be strategically placed to provide effective locomotion while adapting to the bridge's movements.
**3. Weight Distribution:** Keeping the center of mass low and evenly distributing the weight across the robot's body will help maintain balance.
**4. Connection Between Voxels:** Utilizing connections in the matrix to enhance flexibility and adaptability as the bridge deforms.
**5. Combination of Rigid and Soft Voxels:** Using rigid voxels where strength is necessary while incorporating soft voxels for movement and adaptability.

(a) Example of design experience transferred from Walker-v0 to BridgeWalker-v0.

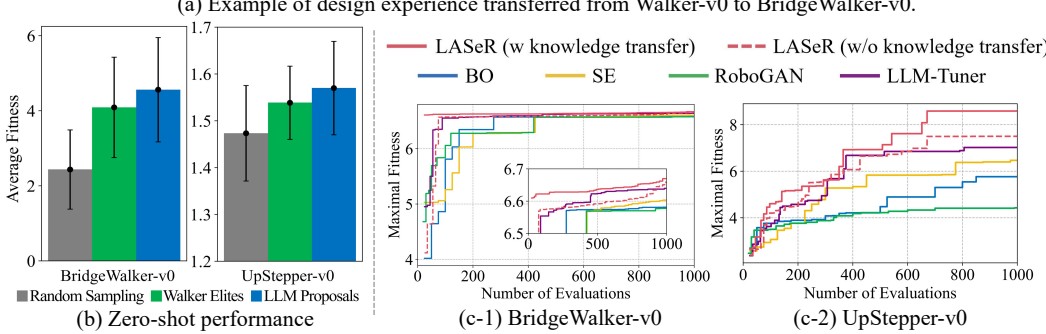

(b) Zero-shot performance  (c-1) BridgeWalker-v0  (c-2) UpStepper-v0

Figure 3: Results of inter-task knowledge transfer.

## 4.3 ABLATION STUDIES

### 4.3.1 EFFECTIVENESS OF DIRECT

As shown in Table 4.3.1, the Diversity Reflection Mechanism fosters a robust increase in diversity compared to an ablated version. It is further demonstrated in Figure 4 that the exploratory behaviors led by DiRect also facilitate more efficient navigating of design spaces, leading to reduced susceptibility to local optima and higher optimization efficiency. These results combine to underscore the distinct superiority of DiRect to yield dual benefits in optimization efficiency and diversity by exploiting the reasoning capabilities of LLMs.

Table 2: Ablative results of diversity (reported as mean (std))

|  | Walker-v0 | Carrier-v0 | Pusher-v0 | Catcher-v0 |
|---|---|---|---|---|
| LASeR | **23.09 (5.33)** | **20.87 (4.27)** | **20.91 (8.85)** | **8.07 (2.48)** |
| LASeR w/o DiRect | 16.96 (1.38) | 7.28 (1.13) | 10.50 (1.04) | 5.65 (1.11) |

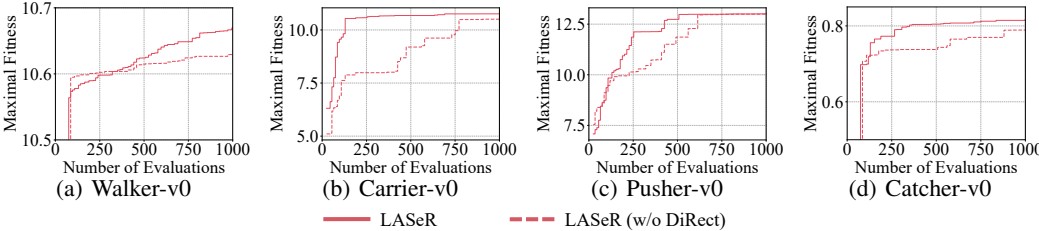

(a) Walker-v0  (b) Carrier-v0  (c) Pusher-v0  (d) Catcher-v0

Figure 4: Ablative results of single-task optimization efficiency. The results of the ablated version are averaged across three repeated runs.

### 4.3.2 EFFECTIVENESS OF TASK-RELATED METADATA

As explained in Section 3.2, current LLM-aided evolutionary frameworks mostly lack sufficient grounding in task-related background information, which potentially impedes their performances in real-world applications. We test this conjecture by removing descriptions of task objectives and simulation environment from our prompts, and see a significant performance drop (Figure 5(a)), hence justifying our prompt design. For finer-grained ablations on individual components of the prompt, please see Appendix K.

### 4.3.3 IMPACT OF LLM VERSION AND TEMPERATURE PARAMETER

Previous work has shown that the temperature parameter of LLMs has an unignorable influence on evolutionary outcomes, with higher temperatures tending to yield better results (Pluhacek et al., 2024). However, we observe a reverse effect where a lower temperature turns out slightly more favorable (Figure 5(b)). We suspect that this is partly due to the complexity within VSR design, which necessitates precise extrapolation from an ascending sequence of solutions. Any deviation could lead to substantial performance drops, outweighing the benefits of random exploration. Meanwhile, we note that the ablation studies with temperature as 1.5 fail similarity checks only about 70% as often as when temperature equals 0.7. In words, higher temperatures would lead to greater but ineffective variability in candidate solutions so that they could bypass diversity reflection. These results suggest that lower output temperatures are required for our approach to work better. Additionally, we observe the same improvement resulted from more up-to-date LLMs as in past literature (Figure 5(c)). This shows promise of robot design automation directly benefiting from better language models, which puts us in a strategic position to ride the wave of rapidly progressing LLMs.

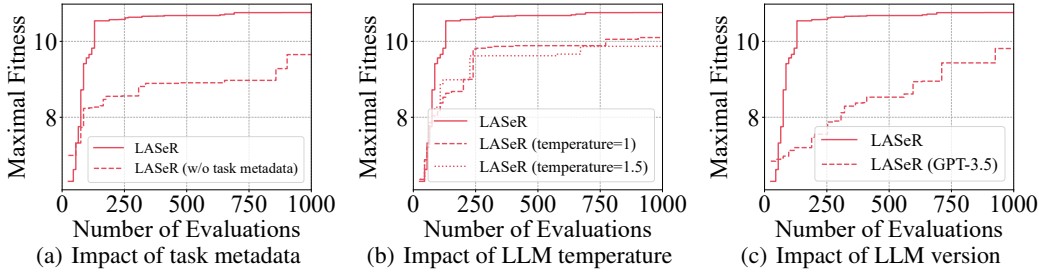

(a) Impact of task metadata     (b) Impact of LLM temperature     (c) Impact of LLM version

Figure 5: Additional ablation studies on Carrier-v0. The results of the ablated versions are averaged across three repeated runs.

## 5 CONCLUDING REMARKS

We present LASeR – Large Language Model-Aided Evolutionary Search for Robot Design Automation. With a novel diversity-oriented reflection mechanism termed DiRect, we elicit intelligent exploratory behaviors from LLMs that reshape the exploration-exploitation tradeoff with dual improvements in optimization efficiency and diversity. We additionally propose to ground robot design on rich task-related metadata and uncover the intriguing inter-task reasoning capabilities of LLMs to foster generalizable design processes across different applications. Our experiments with simulated voxel-based soft robots demonstrate superior performances of our approach compared to competitive baselines. Scaling up LASeR for multi-task optimization would hopefully further harness the inter-task reasoning abilities of LLMs to boost sample efficiency. It is also interesting to investigate how LLM-aided control strategies (Wang et al., 2023a) could be integrated into our framework, so that LLMs are not only responsible for action planning, but also enabled to design their own embodiments, hence exploiting the synergy between design and control. We leave these for our future work. For a more detailed discussion of limitations and open problems, please refer to Appendix N.

### ACKNOWLEDGEMENTS

This work is funded by National Natural Science Foundation of China (No.72371241), the MOE Project of Key Research Institute of Humanities and Social Sciences (22JJD110001), and Zhiqiang Foundation. The authors would like to thank all the anonymous reviewers for their valuable comments.

ETHICS STATEMENT

This work uses simulated task environments which have been commonly used in previous research of robot design automation and should not be regarded controversial. Our use of Large Language Models is strictly confined to simulated robot design generation without real-world deployment, and therefore does not involve any safety risks.

REPRODUCIBILITY STATEMENT

Our code is readily available on GitHub. This work uses GPT-3-turbo and GPT-4o-mini, whose APIs are publicly accessible. However, due to the uncontrollable random generator seeds behind closed-source LLMs, experiments involving these models generally suffer from limited reproducibility (Huang et al., 2024a). Developing reproducible methods for API calls would significantly improve the replicability of research outcomes involving Large Language Models.

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

APPENDIX

# A  FULL PROMPTS

## A.1  SYSTEM PROMPT FOR GENERATING OFFSPRING ROBOT DESIGNS

```
Now you will serve as an intelligent search operator in an Evolutionary Algorithm. In each generation
you are given a number of evaluated solutions in the format of numpy array, together with their fitness
scores. Each solution and its fitness score are separated by a comma. Different solutions are separated
by semicolons. The solutions are sorted according to their fitness scores in ascending order. Higher
fitness scores are better. Your job is to output a new solution that meets a desired fitness. Please
try your best to logically analyze the relationship between the evaluated solutions and their fitness
scores, and adhere to this information while proposing the new solution. A solution is a 5 times 5 matrix,
where each entry is an integer between zero and four. Please begin the new solution with <solution>
and end it with </solution>. The new solution should be formatted in numpy array fashion. The new solution
must be distinct from the evaluated solutions. Only generate the new solution. No explanation.
```

## A.2  USER PROMPT FOR GENERATING OFFSPRING ROBOT DESIGNS (CARRIER-V0 AS AN EXAMPLE)

**## Description of VSRs**
We are going to design the structure of a two-dimensional voxel-based soft robot (VSR) in a simulation
environment. VSRs are composed of square-shaped voxels of different types, aligned into a 5 * 5 matrix.
Adjacent voxels (that is, in either the same row or the same column) are connected together; Voxels
located in diagonal positions are not connected together. The robot is subject to gravity, and the bottom
row touches the ground. There are 5 types of voxels available, including soft voxels (for which elastic
deformation is possible), rigid voxels (which can not deform), horizontal and vertical actuators (which
can change their sizes horizontally or vertically), and empty voxels (which basically mean that the
corresponding position is empty). Empty voxels, rigid voxels, soft voxels, horizontal actuators and
vertical actuators are represented as 0, 1, 2, 3 and 4, respectively.

**## Description of simulation environment**
The simulation represents objects and their environment as a 2D mass-spring system in a grid-like layout,
where objects are initialized as a set of non-overlapping, connected voxels. The simulation converts
all objects into a set of point masses and springs by turning each voxel into a cross-braced square,
which may undergo deformation as the simulation progresses. The springs obey Hooke's law. Note that
adjacent voxels share point masses and springs on their common edge. All point masses in the simulation
have the same mass and the equilibrium lengths of axis-aligned and diagonal springs are constants for
simplicity. However, the spring constants assigned vary based on voxel material-type, with ties broken
in favor of the more-rigid spring. The actuators undergo gradual expansion/contraction either
horizontally or vertically according to action signals, by changing the lengths of the corresponding
springs.

**## Task description**
Your job is to propose robot designs suitable for completing the following task. A three-voxel wide
box is initialized right above the robot, and the robot is required to keep the box on top of its head
stably without letting it slip off, while locomoting rightwards as quickly as possible.

**## Constraints**
There are two constraints to VSR designs: 1. all voxels must form an entirety and should not fall apart;
That is, the four voxels, if any, above, below, to the left and to the right of a non-empty voxel mustn't
be empty at the same time. An example that violates such a constraint is
[[2,2,2,2,2],[1,0,1,0,1],[0,4,3,4,0],[1,3,1,0,1],[0,4,2,4,0]], in which the voxel '1' in the fourth
row and fifth column would fall off because it is not connected to any non-empty voxel; (2) there must
be at least one actuator (that is, either 3 or 4), so that the robot could interact with the environment.

```
## Additional requirements
Please carefully analyze the relationship between evaluated solutions and their fitness scores, and
make use of this information to propose the new solution. Please make use of empty voxels cleverly so
that complex functional substructures could be produced to fulfill the purposes of both carrying and
locomoting. Note that a high-performing robot design is not necessarily symmetric.

(elite design-fitness pairs omitted, sorted in ascending order)

## "Just-ask" query
Now please generate a new robot design that has a fitness of {str(1.2*current_max_fitness)}.
```

## A.3    USER PROMPT OF DIRECT

```
The solution that you just generated is too similar with an existing one. It needs further modification
to improve diversity. Please decide which voxels in the solution can be replaced by other types of materials,
without harming its fitness score. Change no more than 3 voxels. Please base your analysis on the
characteristics of the evaluated solutions given to you. Meanwhile, make sure that the modification
does not violate the constraints of VSRs. Now please tell me which voxels exactly do you think can be
altered, and explain the reason.

(LLM suggesting modifications)

Based on your analysis above, please generate the resulting solution. The number of voxels changed should
not exceed three. Do not provide further texual explanation.
```

## A.4    USER PROMPT FOR INTER-TASK KNOWLEDGE TRANSFER (WALKER-V0 →
###    BRIDGEWALKER-V0 AS AN EXAMPLE)

```
(Descriptions of VSRs, simulation environment and constraints are same as those in A.1 and therefore
omitted. )

We already have some high-performing robot designs from a task named 'Walker', where the robot is required
to locomote rightwards as quickly as possible on rigid flat terrain. The robot designs are as follows:

(Elite Walker-v0 designs omitted. )

Now your job is to propose ten high-performing robot designs for another task named 'BridgeWalker',
where the robot is required to locomote rightwards as quickly as possible on a soft rope-bridge rather
than rigid flat terrain. Please analyze the potential correlation between the two tasks, identify the
knowledge that can be transferred from Walker to BridgeWalker, and give ten robot designs that are suitable
for BridgeWalker. Give your answer in numpy array fashion. Enclose each design between <Solution> and
</Solution>.
```

# B    INTRODUCTION TO TASK INSTANCES

In this section, we briefly introduce the tasks we adopted for benchmarking in Evolution Gym. This introduction is heavily borrowed from their original paper (Bhatia et al. 2021).

We first define some notations that would be used later.

- **Position:** Denote with $p^o$ the position of the center of mass of an object $o$, which consists of two components $p^o_x$ and $p^o_y$, *i.e.*, the positions on $x$ and $y$ axis. $p^o$ is derived by averaging the positions of all the point-masses that make up object $o$;
- **Velocity:** Denote with $v^o$ the velocities of the center of mass of an object $o$, which consists of two components $v^o_x$ and $v^o_y$, *i.e.*, the velocity on $x$ and $y$ axis. $v^o$ is computed by averaging the velocities of all point masses that make up object $o$;

- **Orientation:** Denote with $\theta^o$, a vector of length one, the orientation of an object $o$. Denote the position of point mass $i$ of object $o$ as $p_i$, and $\theta^o$ is computed by averaging over all $i$ the angle between the vector $p_i - p^o$ at current time and the initial state. This average is weighted by $||p_i - p^o||$ in the initial state.

- **Other observations:** Let $c^o$ be a vector of length $2n$ that describes the relative positions of all $n$ point masses of object $o$ to the center of mass. Let $h_b^o(d)$ characterize the terrain information around a robot below its center of mass. More specifically, for some integer $x \leq d$, the corresponding entry in vector $h_b^o(d)$ will be the highest point of the terrain which is lower than $p_y^o$ between a range of $[x, x+1]$ voxels from $p_x^o$ in the $x$-direction.

- Besides, we would denote the robot as object $r$, the box that it is trying to manipulate as object $b$, the number of point masses in $r$ as $n$, the observation vector as $\mathcal{S}$, and the reward function as $R$.

## B.1 WALKER-V0

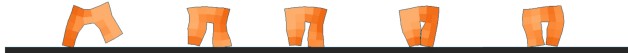

Figure 6: Walker-v0

In this task, the robot is required to walk as far as possible on flat terrain. $\mathcal{S} \in \mathbb{R}^{n+2}$ consists of $v^r$ and $c^r$ with lengths 2 and $n$. $R = \Delta p_x^r$ rewards the robot for moving in the positive $x$-direction. The robot is also given a one-time reward of 1 for reaching the end of the terrain.

## B.2 CARRIER-V0

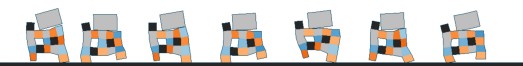

Figure 7: Carrier-v0

In this task, the robot is required to catch a box initialized above it and carries it as far as possible. $\mathcal{S} \in \mathbb{R}^{n+6}$ consists of $v^b$, $p^b - p^r$, $v^r$ and $c^r$ with lengths 2, $n$, 2 and 2 respectively. $R = R_1 + R_2$, where $R_1 = 0.5 \cdot \Delta p_x^r + 0.5 \cdot \Delta p_x^b$ rewards the robot and the box for moving in the positive $x$-direction, and $R_2 = 0$ if $p_y^b \geq t_y$ and otherwise $10 \cdot \Delta p_y^b$ penalizes the robot for dropping the box below a threshold height $t_y$. The robot is also given a one-time reward of 1 for reaching the end of the terrain.

## B.3 PUSHER-V0

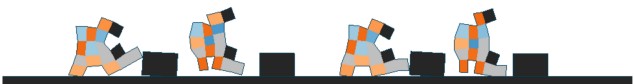

Figure 8: Pusher-v0

In this task, the robot is required to push a box initialized in front of it. $\mathcal{S} \in \mathbb{R}^{n+6}$ consists of $v^b$, $p^b - p^r$, $v_r$ and $c^r$ with lengths 2, $n$, 2 and 2 respectively. $R = R_1 + R_2$, where $R_1 = 0.5 \cdot \Delta p_x^r + 0.75 \cdot \Delta p_x^b$ rewards the robot and the box for moving in the positive $x$-direction, and $R_2 = -\Delta|p_x^b - p_x^r|$ penalizes the robot and the box for separating in the $x$-direction. The robot is also given a one-time reward of 1 for reaching the end of the terrain.

### B.4 CATCHER-V0

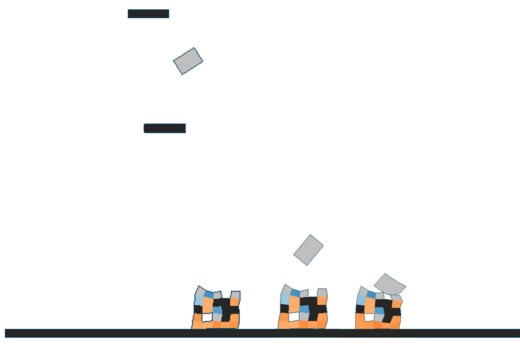

Figure 9: Catcher-v0

In this task, the robot is required to catch a fast-moving and rotating box. The observation space $\mathcal{S} \in \mathbb{R}^{n+7}$ consists of $p^b - p^r$, $v^r$, $v^b$, $\theta^b$ and $c^r$, with lengths 2, 2, 2, 1 and $n$, respectively. The reward $R = R_1 + R_2$, where $R_1 = -\Delta|p_x^b - p_x^r|$ rewards the robot for approaching the box in the $x$-direction, and $R_2 = 0$ if $p_y^b \geq t_y$ and $10 \cdot \Delta p_y^b$ otherwise penalizes the robot for dropping the box below a threshold height $t_y$.

### B.5 BRIDGEWALKER-V0

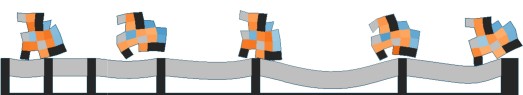

Figure 10: BridgeWalker-v0

In this task, the robot is required to walk as far as possible on a soft rope-bridge. $\mathcal{S} \in \mathbb{R}^{n+3}$ consists of $v^r$, $\theta^r$ and $c^r$ with lengths 2, 1 and $n$ respectively. $R = \Delta p_x^r$ rewards the robot for moving in the positive $x$-direction. The robot is also given a one-time reward of 1 for reaching the end of the terrain.

### B.6 UPSTEPPER-V0

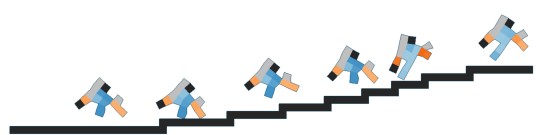

Figure 11: UpStepper-v0

In this task, the robot is required to mount stairs of varying lengths. $\mathcal{S} \in \mathbb{R}^{n+14}$ consists of $v^r$, $\theta^r$, $c^r$ and $h_b^r(5)$ with lengths 2, 1, $n$ and 11, respectively. $R = \Delta p_x^r$ rewards the robot for moving in the positive $x$-direction. The robot is given a one-time reward of 2 for reaching the end of the terrain, and a one-time penalty of -3 for rotating more than 75 degrees from its original orientation in either direction (after which the environment is reset).

## C  HYPERPARAMETER SETTINGS

Table 3: Hyperparameter settings

| hyperparameter | value |
|---|---|
| LASeR | |
| robot size (height $\times$ width) | $5 \times 5$ |
| upper limit of LLM interactions | 200 |
| probability of similarity check in DiRect | 0.4 |
| similarity threshold in DiRect | no more than 20 identical voxels |
| percentage of survivors in natural selection | linearly decreasing from 60% to 8% |
| PPO Policy Training | |
| number of parallel sampling processes | 4 |
| number of time steps in each process | 128 |
| learning rate | $2.5 \times 10^{-4}$ |
| $\epsilon$ in the clip function of PPO | 0.1 |
| number of iterations | 1000 |
| number of epochs per iteration | 4 |
| number of mini-batches per epoch | 4 |
| $\lambda$ in generalized advantage estimation (GAE) | 0.95 |

## D  RESULTS OF SIGNIFICANCE TESTS

We conduct significance tests between LASeR and LLM-Tuner, the most competitive baseline. Since in this work we have chose a sufficiently large number of robot evaluations (*i.e.,* 1000) to give ample opportunity to all algorithms to converge, it becomes more relevant to compare the convergence speed rather than entire fitness curves. To this end, we first average the eventual fitness values obtained by all repeated experiments (denoted as $f$) within a given task, and then record the number of evaluations that each experiment took to reach this average fitness (denoted as $n$). For those that did not reach $f$, $n$ is simply recorded as 1000. We then conduct a two-tailed $t$-test to compare the $n$'s of different algorithms. For Carrier-v0, $f$ is 10.69, and $n$ is on average **719.2** and **979** for LASeR and LLM-Tuner, respectively ($p =$**0.029**). For Pusher-v0, $f$ is 12.95, and $n$ is on average **528** and **888.6** with $p =$**0.054**. For Walker-v0, since none of the experiments of LLM-Tuner reach $f$=10.65, we instead compare the eventual fitness values achieved by LASeR and LLM-Tuner, which are on average **10.67** and **10.63**, with $p <$**0.001**. For Catcher-v0, the eventual fitness values of LASeR and LLM-Tuner are **0.816** and **0.802**, with $p$=**0.08**.

## E  COMPARISON OF DIRECT AND RANDOM VOXEL EDITING

To further showcase the effectiveness of our diversity reflection mechanism (DiRect), we re-implemented our experiments with DiRect replaced by random voxel mutations. Specifically, we implemented random editing by substituting the diversity reflection (DiRect) mechanism with random voxel mutation, which is supported by a built-in function of EvoGym. Specifically, we found that the number of voxels edited by DiRect in each design is about **2.61** on average. Thus, for random mutation, we set the mutation rate to be **0.1**, *i.e.,* each voxel will, with a probability of 0.1, be randomly replaced by a different material. Given that a robot design consists of 25 voxels, this results in **2.5** voxels being edited on average, which we believe is reasonably close to DiRect editing.

We conducted a paired two-tailed Student's $t$-test, and found that the fitnesses of randomly mutated robot designs are significantly lower than their pre-editing counterparts ($p < 0.001$). In contrast, the fitnesses of robot designs before and after DiRect modification show no significant difference ($p = 0.19$). These results suggest that stochastic exploratory behaviors would disrupt essential functional structures, whereas LLM-aided diversity reflection holds a distinct "***informed***" nature. Specifically, it builds on successful designs discovered along the evolutionary trajectory, and promotes exploration without compromising functionality.

For illustrative purposes, here we present two examples of diversity reflection, including both the pre- and post-editing morphologies, along with explanations provided by the LLM for its modifications. It can be seen from Figure 12 that the LLM is indeed able to identify critical substructures within robot designs, and modifies only the voxel placements that do not affect functionality, yet promote diversity. These results provide sound evidence that DiRect is reliably functioning as an ***intelligent* mutation operator**.

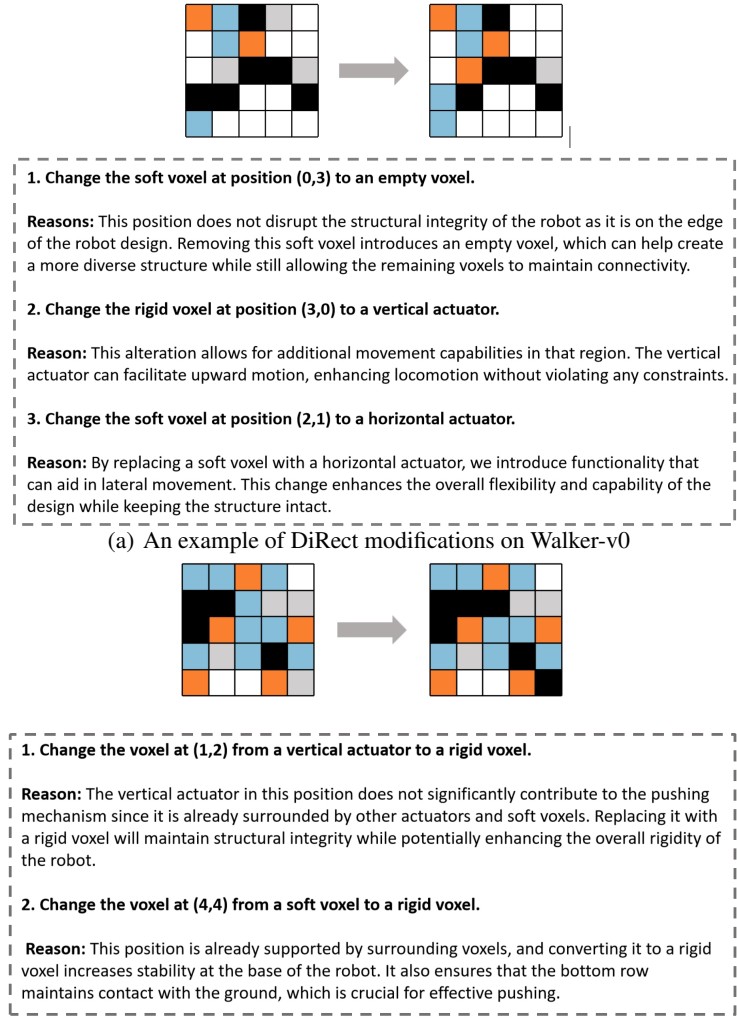

(a) An example of DiRect modifications on Walker-v0

(b) An example of DiRect modifications on Pusher-v0

Figure 12: Illustrative examples of DiRect modifications. Each example includes the pre- (left) and post- (right) editing designs, together with the modifications and justifications provided by an LLM to enhance morphological diversity.

## F    COMPARISON WITH TWO ADDITIONAL BASELINES

We conducted comparative studies between LASeR and two additional baseline algorithms. The first one is OPRO (Yang et al., 2024), another evolutionary strategy that uses LLMs as search operators, which we adapted for voxel-based soft robot (VSR) design. The second one is MorphVAE (Song et al., 2024a), a state-of-the-art co-design algorithm that does not employ LLMs but is also developed on the EvoGym platform.

As shown in Figure 13, LASeR consistently outperforms the two baselines in terms of optimization efficiency, reflected by its steeper fitness curves. Here we would like to clarify that we have deliberately chosen a **sufficiently large** number of robot evaluations to hopefully allow all algorithms

to converge for fair comparison. This explains why different algorithms end up with rather similar fitness levels. However, in the context of robot design automation, the convergence rate is an important aspect for evaluating design algorithms, as the evaluation of robot designs usually involves computationally expensive control learning, let alone the manufacturing costs of physical robots when deployed in real-world application. In this regard, LASeR exhibits considerable performance gains.

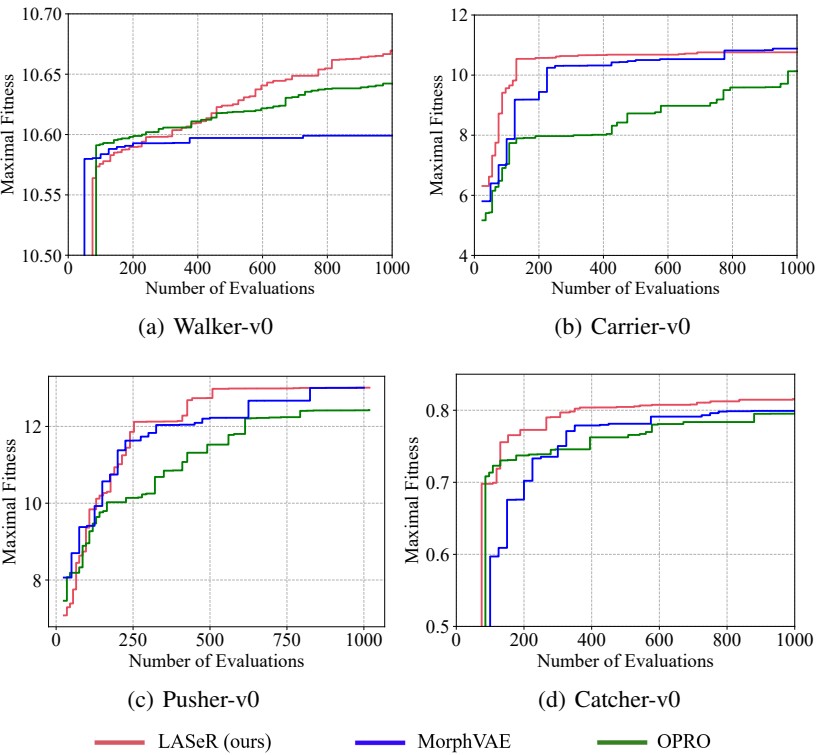

(a) Walker-v0             (b) Carrier-v0

(c) Pusher-v0             (d) Catcher-v0

— LASeR (ours)     — MorphVAE     — OPRO

Figure 13: Comparison between LASeR (red), MorphVAE (blue) and OPRO (green) in terms of optimization efficiency. In the original paper of MorphVAE (Song et al., 2024a), two variants are proposed that focus on either optimization efficiency or diversity. We conduct three repeated trials for each of them and take an average to reflect the overall performance of MorphVAE. This is also the case for diversity calculation.

Furthermore, as demonstrated in Table 4, LASeR achieves the highest diversity in two out of four tasks, while MorphVAE is dominant in the remaining two tasks. Despite the competitive performance of MorphVAE, we note that LASeR holds distinct advantages: (a) with the novel diversity reflection mechanism, LASeR is capable of achieving a **more favorable trade-off** between optimization efficiency and diversity, whereas MorphVAE proposes two variants, each of which focuses on one aspect and compromises the other; (b) MorphVAE leverages a variational autoencoder to approximate the high-performing robot distribution and generate offspring solutions, which lacks interpretability. On the contrary, LASeR can instruct an LLM to explicitly explain its design choices and thus provide valuable insights for robot design (see Appendix H). LASeR is also capable of more intelligent knowledge transfer across different tasks, utilizing the reasoning capabilities of LLMs. Furthermore, the inferior diversity outcomes of OPRO once again reveal the inefficiency of LLMs to balance exploitation with exploration on their own without diversity reflection.

## G   EVALUATION ON 10x10 WALKER-V0

In this work, we adhered to the standard setup used in previous VSR studies, specifically a 5x5 body size with five different materials, as this configuration is proven already expressive enough for complex and diverse morphological structures to emerge (Song et al., 2024a; Saito & Oka, 2024;

Table 4: Comparison between LASeR, MorphVAE and OPRO in terms of diversity.

|  | Walker-v0 | Carrier-v0 | Pusher-v0 | Catcher-v0 |
|---|---|---|---|---|
| MorphVAE | 16.20 (N/A) | **33.16 (16.59)** | 18.18 (12.48) | **11.00 (3.09)** |
| OPRO | 18.95 (4.19) | 6.01 (2.41) | 9.42 (2.08) | 6.41 (1.48) |
| **LASeR (ours)** | **23.09 (5.33)** | 20.87 (4.27) | **20.91 (8.85)** | 8.07 (2.48) |

Dong et al., 2023; Wang et al., 2023a;b). Nevertheless, to evaluate the scalability of our approach to larger design spaces, we tested both LASeR and LLM-Tuner on Walker-v0 with a 10x10 body size. Our findings, as presented in Figure 14, demonstrate that LASeR continues to outperform the baseline in terms of optimization efficiency, even in this larger design space. We attribute this success to the unique advantage of LLMs. Specifically, LLMs leverage their reasoning capabilities to identify favorable voxel assembly patterns within high-performing designs, instead of relying on random mutations (as seen in genetic algorithms and other heuristics), to generate offspring solutions. This is also demonstrated in Appendix E and H, where we show that LLMs are able to provide justifications for their decision making when generating offspring solutions and carrying out diversity reflection. It is worth noting that **the 10x10 configuration results in a design space that is** $2.65 \times 10^{52}$ **times larger than the 5x5 case**, due to combinatorial explosion. Therefore, the promising results indicate a remarkable potential of LLM-based evolutionary strategies to scale to even larger and more complex robot design problems.

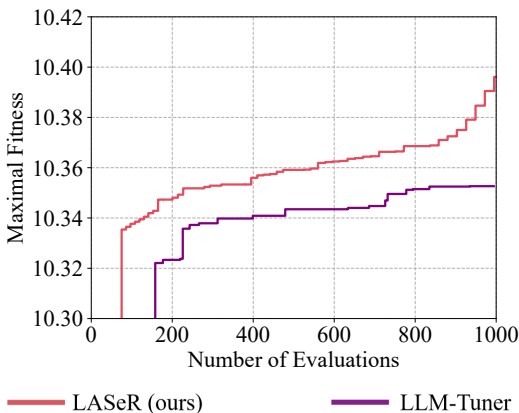

Figure 14: Comparison of LASeR (red) and LLM-Tuner (purple) on 10x10 Walker-v0. The results are averaged across four independent runs.

## H  LLM's Interpretation of Robot Design

In our preliminary experiments, we included explicit LLM reasoning, where the LLM was prompted to explain its design choices (similar to chain-of-thought). However, we observed no significant performances gains by doing so, and therefore opted to remove the reasoning process to speed up evolution and save API costs. Nevertheless, we would like to clarify that our approach is capable of affording higher interpretability. This can readily be achieved by explicitly instructing the LLM to explain its decision-making process, rather than serving as a black box. We provide in Figure 15 the output generated by GPT-4o-mini after receiving a collection of high-performing Catcher-v0 designs and instructed to generate a new one with higher performance. The explanations provided are insightful and reasonable, revealing the advantageous structures present in high-performing designs. This suggests that LLMs are indeed able to align specific voxel assembly patterns with high performance, and leverage these principles to generate improved offspring solutions.

The designs provided demonstrate several key principles that contribute to their performance in the Catcher-v0 task, where the goal is to catch and carry a box. Here's a summary of observed design principles:

**1. Structural Stability:** The bottom row is consistently filled with rigid and soft voxels, which provide a solid foundation and enhance stability when carrying loads. Rigid voxels are typically positioned towards the bottom or at critical structural points to resist deformation under load.

**2. Effective Use of Actuators:** The designs incorporate both horizontal and vertical actuators strategically. This allows for dynamic adjustments to the robot's shape to better engage with the carried box. The placement of actuators near the center of mass or at junctions of flexible structures facilitates effective load distribution and manipulation.

**3. Balance of Soft and Rigid Voxels:** A mix of soft and rigid voxels provides the flexibility needed for absorbing shocks while also maintaining structural integrity. Designs often use soft voxels in upper layers to allow for some deformation without compromising overall structure, which can help absorb impacts when catching the box.

**4. Optimized Catching Mechanism:** Successful designs tend to have a specific configuration that aids in maximizing the area available for catching the box. This often involves using soft voxels around the central area to create a "cushion" effect. The presence of empty voxels strategically placed helps optimize the weight distribution and maneuverability.

**5. Symmetry and Connectivity:** Many designs exhibit symmetrical features, which can help ensure balanced weight distribution and stability during movement. The connectivity between adjacent voxels supports resilience against shear forces and promotes coordinated movements, particularly important during the catch-and-carry process.

**6. Adaptability:** The ability to dynamically adjust through the actuators enables the robot to adapt to various scenarios—this is critical when dealing with different box sizes or weights.

Here's a proposed new catcher design aimed at enhancing proficiency in the task:
proposed_design = np.array([[3, 4, 0, 2, 1],[1, 4, 3, 1, 3],[3, 2, 4, 3, 4],[4, 1, 2, 3, 1], [0, 1, 2, 1, 2]])
**Design Features:**

**1. Enhanced Stability:** The bottom row consists of two rigid voxels flanking a soft voxel providing a stable base while allowing slight deformation.

**2. Improved Actuation:** The second row contains two actuators (one vertical and one horizontal), enabling more versatile manipulations and adaptations to different box sizes.

**3. Strategic Placement of Soft Voxels:** The arrangement of soft voxels in the middle provides cushioning during the catch phase, helping to absorb any sudden movements.

**4. Central Weight Distribution:** The design emphasizes centralized soft voxels, allowing for effective engagement and support of the box once caught.

**5. Flexibility and Adaptation:** Additional soft voxels above the actuators ensure adaptability in height and width as needed, improving the ability to secure and carry the box efficiently.

This proposed design aims to leverage the advantages of existing high-performing designs while introducing modifications that could enhance its capacity to perform well in the 'Catcher-v0' task. By emphasizing stability, adaptability, and an effective actuation mechanism, this design could potentially outperform prior iterations.

Figure 15: The LLM-generated output on Catcher-v0. In this example, the LLM is prompted to explain its decision-making process when generating a new robot design.

## I  VISUALIZATION OF EVOLVED ROBOT DESIGNS

Here we provide visualizations of robot designs evolved by LASeR and baseline algorithms. We take Pusher-v0 as an example, and randomly select five high-performing robot designs obtained by each algorithm in a single run of experiment. It can be seen from Figure 16 that the robot designs evolved by LASeR seem to exhibit the highest level of variability. However, we would like to make two notes here. First, as we are only able to display a small fraction of robot designs, this visualization could only serve as a qualitative and intuitive verification of results presented in Section 4.2.1, rather than a valid means of evaluation on its own. Second, as we would detail in Appendix J, the number of evolved high-performing designs should also be taken into account when measuring morphological

diversity. To this end, we believe the quantitative results reported in Section 4.2.1 still serve as the most comprehensive and reliable evaluation of diversity.

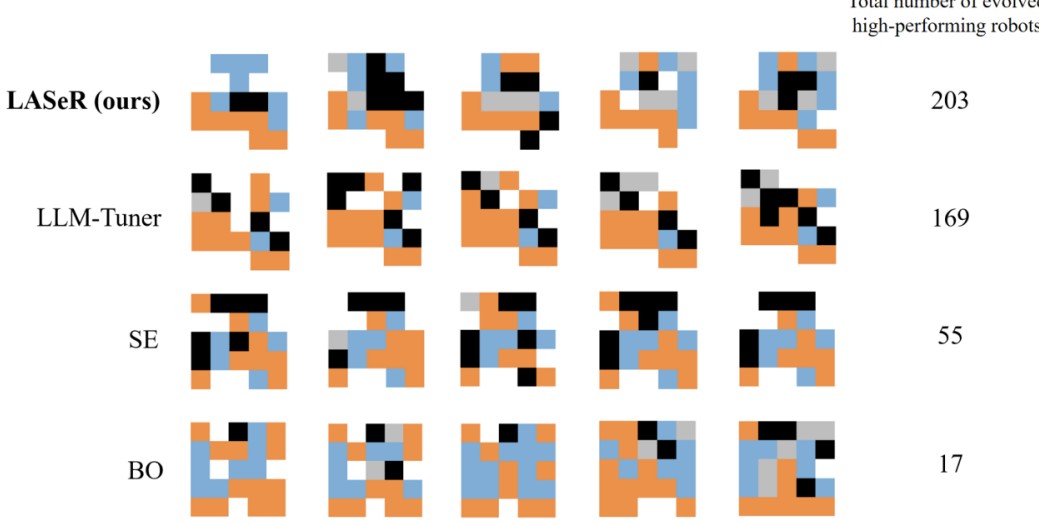

Figure 16: Visualizations of evolved high-performing robot designs on Pusher-v0. For each algorithm, five designs are randomly chosen for display. Annotated on the right are total numbers of high-performing designs obtained. The result of RoboGAN is absent because it fails to obtain robot designs that surpass the fitness threshold (*i.e.,* the 90% quantile of fitnesses achieved by all algorithms).

## J FURTHER DISCUSSION ON DIVERSITY MEASUREMENT

Diversity is an important aspect for evaluating robotic systems and, in turn, the performance of robot design algorithms, as diversified design alternatives are crucial for handling dynamic environments and increasing robustness of robotic systems. To our knowledge, previous studies have predominantly employed two methods for quantifying diversity: (a) averaged measures of distinctiveness within a group of robots, such as per-voxel entropy (Song et al., 2024a) and pair-wise edit distance (Saito & Oka, 2024); (b) manual categorization of robot designs into distinct classes, followed by the calculation of the Simpson index, which is analogous to an entropy measure of class distribution (Medvet et al., 2021). The latter method becomes impractical when dealing with more abstract morphologies without clear subpopulations. The former, on the other hand, presents a **paradox** (Figure 17): including a new robot design into an existing collection can reduce diversity, even if the new design is distinct, provided that it falls within the distribution of this collection. Here, by "falling within the distribution" we mean that the distance between the new design and existing ones is on average smaller than that within the original collection.

To address the above issue, we incorporate the number of distinct robot designs into our measurement as a *correction*. Thus, our two measures – **edit distance** (measuring the distinctiveness of evolved designs) and **the number of distinct designs** – complement each other, providing a **more comprehensive and reasonable** characterization of diversity. However, we acknowledge that the weights assigned to these quantities (1.0 and 0.1) are somewhat expedient and primarily intended to bring them onto the same scale. This is based on our preliminary experiments where we found that the number of distinct high-performing designs obtained in a single run of experiment typically ranged from several dozens to around two hundred, while the edit distance is defined to range between 0 and 25. Given the lack of universally accepted metrics for measuring morphological diversity, we hope we could inspire future work to devise even more reasonable and comprehensive approaches.

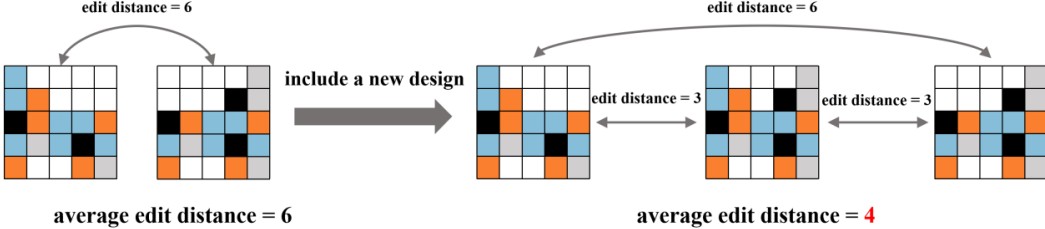

Figure 17: A paradox with diversity measurement. The inclusion of a new, distinct robot design decreases, rather than increases, the diversity when solely measured as the edit distance. This is counter-intuitive as the addition of a distinct alternative should benefit diversity.

We additionally provide the two separate measures of diversity before being weighted averaged (Table 5 and 6). Our finding suggests that LASeR has more of a advantage in discovering distinct high-performers than achieving high averaged edit distance. We cannot actually state which approach is more favorable, as both benefit diversity in their own way. However, combining the results of optimization efficiency (*i.e.,* the fitness curves), it is clear that LASeR better balances exploration with exploitation.

Meanwhile, as we pointed out above, we believe that the results in Table 6 are somewhat misleading, because edit distance in itself does not suffice as a valid diversity metric. Even if a group of robot designs has another group as its subset, the former might still have a lower edit distance (even much lower, due to the paradox in Figure 17). The comparative results in Table 6 might not be in our favor, but we choose to display them to reveal **an open problem regarding diversity measurement** and hopefully inspire future work to investigate further.

Table 5: Comparative results of the number of distinct high-performing designs.

|  | Walker-v0 | Carrier-v0 | Pusher-v0 | Catcher-v0 |
|---|---|---|---|---|
| BO | N/A | 22.00 (6.38) | 13.33 (3.86) | 3.67 (1.25) |
| SE | 13.25 (3.03) | 34.40 (26.62) | 6.00 (3.85) | 7.00 (2.83) |
| RoboGAN | N/A | 3.00 (N/A) | N/A | 3.00 (0.82) |
| LLM-Tuner | 33.75 (41.54) | 55.00 (67.02) | 46.25 (70.90) | 5.67 (1.70) |
| MorphVAE | 2.00 (N/A) | **241.17 (179.83)** | 101.50 (133.73) | 11.00 (10.18) |
| OPRO | 138.00 (31.73) | 17.75 (16.04) | 53.25 (20.58) | 9.50 (2.29) |
| **LASeR(ours)** | **165.80 (53.09)** | 162.00 (43.23) | **158.40 (88.68)** | **28.00 (23.69)** |

Table 6: Comparative results of average edit distance.

|  | Walker-v0 | Carrier-v0 | Pusher-v0 | Catcher-v0 |
|---|---|---|---|---|
| BO | N/A | 9.68 (2.25) | **12.35 (2.48)** | **19.39 (0.98)** |
| SE | 3.92 (0.46) | 11.82 (2.66) | 6.66 (4.03) | 13.21 (2.01) |
| RoboGAN | N/A | 10.64 (N/A) | N/A | 17.97 (1.5) |
| LLM-Tuner | 8.23 (1.79) | **12.76 (1.69)** | 9.54 (3.42) | 13.33 (7.21) |
| MorphVAE | **16.00 (N/A)** | 9.04 (3.36) | 8.03 (3.72) | 9.90 (4.00) |
| OPRO | 5.15 (1.02) | 4.24 (1.46) | 4.10 (0.39) | 5.46 (1.30) |
| **LASeR(ours)** | 6.51 (0.09) | 4.67 (0.34) | 5.07 (0.43) | 5.27 (0.31) |

## K  FINER-GRAINED ABLATION ON PROMPT DESIGN

The prompt used in our study consists of three major components: task-related metadata, elite design-fitness pairs, and target fitness.

- The **task-related metadata** primarily includes descriptions of task objectives and the simulation environment. This component is largely derived from the official documents of EvoGym (Bhatia et al., 2021), with **minimal modifications**. This metadata, which is often overlooked in previous works on LLM-aided robot design, serves two main purposes: to ground the evolutionary process in the specific context of the problem, and to facilitate the transfer of knowledge between different tasks.

- The second component consists of **elite design-fitness pairs** previously evaluated, where the designs are sorted according to their fitness in ascending order. This sorting is intended to leverage the pattern-completion capabilities of LLMs, a technique shown to be effective in prior research (Lange et al., 2024; Yang et al., 2024).

- The third component, the **target fitness** (also referred to as the "just-ask query" in Lim et al. (2024)), is introduced as a means of aligning the LLM's outputs with our desired results.

We have demonstrated the indispensability of task-related metadata in Section 4.3.2. To further justify our prompt design and to complement the intuitive explanations provided above, we conducted finer-grained ablation studies and the results are reported in Figure 18. Specifically, we remove the following components one at a time: (a) the description of the simulation engine; (b) the description of task objectives; (c) the just-ask query (or target fitness); in this case, the LLM is simply prompted to generated robot designs with *higher* fitness; and (d) the ascending ordering of elite design-fitness pairs according to fitness. Our findings suggest that removing any of these components leads to performance drops. The just-ask query is proven the most essential, while simulation description and ordering play less important roles.

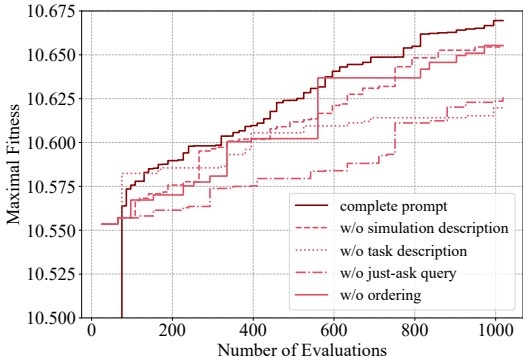

Figure 18: Finer-grained ablation studies on prompt design.

We would like to note that the phrasing of these components is intentionally left simple and intuitive, without applying special techniques of prompt engineering. As such, our experimental results possess a certain level of **robustness** and do not hinge on the specifics of prompt designs. However, it would be a promising direction to integrate various prompting techniques into our framework for better performances.

## L  FURTHER DISCUSSION ON THE SIMILARITY THRESHOLD

In our diversity reflection mechanism (DiRect), a similarity threshold is needed to decide whether a newly generated robot design is overly similar to existing ones and therefore should undergo modifications by diversity reflection. This threshold is indeed a crucial hyper-parameter that controls the performance of LASeR. The choice of this threshold reflects the extent of diversity that one expects to see in the evolved solutions, and hence should be driven by the user's specific preferences.

For instance, setting it as 20 (as we did in our experiments) means that if a newly generated design shares more than 20 identical voxels with any existing solution, it will be modified by DiRect to introduce more variability.

Here, we present some general principles for choosing this parameter. These principles are supported by our additional experiments with several different values of threshold (as shown in Figure 19). High similarity thresholds, like 23, are generally not recommended, as they would hinder the beneficial exploration enabled by LLMs. Conversely, excessively low thresholds (such as 15) might increase diversity but also risk overly aggressive exploration that compromises functionality and, in turn, harms optimization efficiency. We believe this is due to the poor extrapolation performance of LLMs when required to propose robot designs that are much different from given examples. Any moderate values in the middle should lead to desirable performances. In fact, our findings suggest that a threshold of 18 leads to further performance gains beyond 20. However, we note that lower thresholds also more frequently trigger DiRect, which means more LLM API calls. Hence, the threshold choice also involves a trade-off between evolutionary performance (including both optimization efficiency and diversity) and computational costs, and should be considered case by case. We believe adaptive threshold scheduling, based on problem specifics and evolutionary outcomes, could be a promising direction for future research.

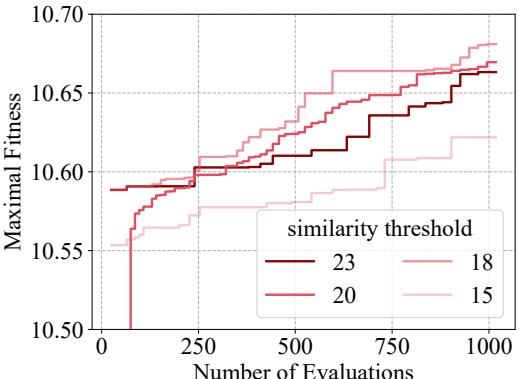

Figure 19: Additional experiments with several different similarity thresholds. The diversity results of these experiments are N/A, 28.51, 23.09, and 13.46 in the order of 15, 18, 20 and 23. The diversity is not available for 15 because the corresponding experiment failed to generate more than one high-performing robot designs.

## M   GENERALIZATION ACROSS DIFFERENT TASKS

In Section 4.2.2, we focused on transferring design experience between task instances that are intuitively similar (*i.e.,* Walker-v0, BridgeWalker-v0 and UpStepper-v0). These experimental designs are largely based on the structural similarities between tasks as revealed in Wang et al. (2023b). Here we demonstrate that this prior knowledge of task relationships is not strictly necessary for successful inter-task generalization. Specifically, we performed an additional zero-shot design experiment (Figure 20) where the LLM was given elite samples of Walker-v0, but instructed to propose designs for a significantly different task, Jumper-v0, in which the robot is required to jump as high as possible. Despite the discrepancy between task objectives, we find that the LLM is able to dig deeper into the underlying inter-task associations and identify rather general, low-level design principles that are still relevant to the new task. Moreover, the zero-shot designs generated by LLM still outperform randomly sampled ones. See Figure 21 for visualizations of some elite Walker-v0 designs, together with a Jumper-v0 design generated by LLM. These findings suggest that LLMs possess substantial potential for generalizing experience across seemingly different optimization problems, as long as they are not completely irrelevant to one other.

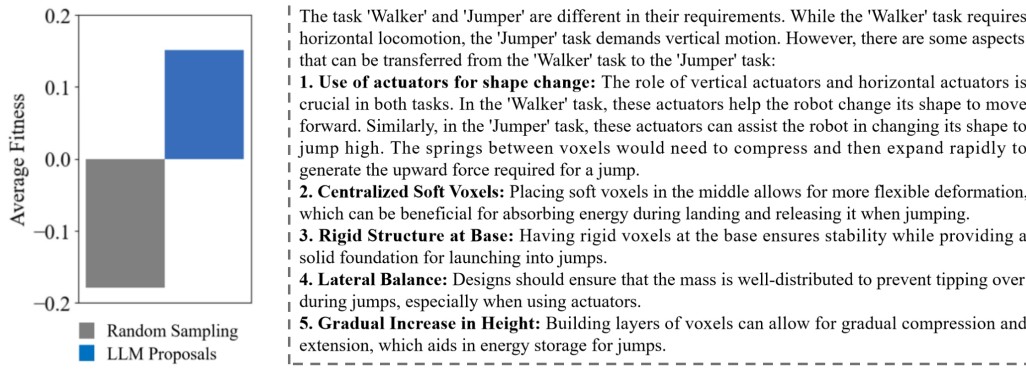

(a) Zero-shot performance (b) Design principles that LLM generalizes from Walker-v0 to Jumper-v0

Figure 20: Additional zero-shot design experiment on Jumper-v0. The zero-shot performance is averaged over 10 robot designs for both random sampling and LLM proposals. Despite the task discrepancy, the LLM is still able to identify general principles, such as the importance of actuators for shape change and rigid voxels for stability, to transfer to the new task.

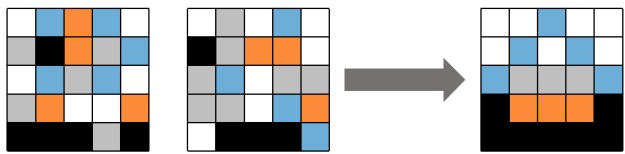

Some of the elite Walker-v0 designs         An LLM-proposed Jumper-v0 design

Figure 21: A visualization of some elite Walker-v0 designs (left), and the Jumper-v0 design generated by LLM (right). The Jumper-v0 design seems different than its Walker-v0 counterparts, but they still share some basic design principles, such as rigid voxels in the bottom row, soft voxels in the middle rows and well-distributed mass. These features largely mirror the analysis by LLM in Figure 20.

## N  FURTHER DISCUSSION ON LIMITATIONS AND FUTURE WORK

Here we expand the discussion of limitations within this work, and outline several open problems that we find promising for future research.

- In this work, we developed our approach on voxel-based soft robots (VSRs), and adhered to the 5x5 body size commonly adopted in previous VSR studies. However, we would like to note that our approach is conceptually adaptable to larger design spaces (as verified in Appendix G) and potentially other types of robots. We look forward to seeing our LLM-based evolutionary framework evaluated in more extensive and complex design problems.

- Regarding inter-task generalization, we focused on intuitively similar tasks in our paper. However, we proved that this prior knowledge of task relationships is not necessary for successful transfer of design experience (as demonstrated in Appendix M). It would be promising to investigate how this inter-task reasoning capability of LLMs could be leveraged in a broader range of optimization problems.

- Currently, we employed LLMs in the most cost-effective way for robot design automation, that is, through in-context learning. It remains to be studied whether fine-tuning LLMs could lead to further performance gains. To our knowledge, fine-tuning Large Language Models typically involves substantially higher costs, both in terms of computational resources and the need of a sufficiently large and carefully curated dataset. However, performance might not be guaranteed due to issues like overfitting and catastrophic forgetting.

Nevertheless, the prospect of a general-purpose LLM fine-tuned for various combinatorial optimization problems is intriguing, and represents an open problem for future research.

- In this work, we followed the standard control learning approach used in previous EvoGym-based studies, *i.e.,* PPO algorithm with MLPs serving as control networks. Since our primary focus is on the evolutionary capabilities of LLMs, we chose this simple yet effective approach for fitness evaluation. That being said, a broad range of alternatives, such as value-based methods and differential simulations (Strgar et al., 2024; Cochevelou et al., 2023), could be tried out, with their impact on LLM-aided evolution examined. We also find it an interesting direction to integrate LLM-aided control strategies (such as those in Wang et al. (2023a), Brohan et al. (2023) and Cheang et al. (2024)) into our framework, so that LLMs are capable of both designing and controlling their own embodiments.

- Morphological diversity is an important aspect for evaluating robot design algorithms, as diversified designs are crucial for ensuring the robustness of robotic systems in highly volatile environments. We pointed out the limitations of previous diversity measures in Appendix J and proposed to make a correction by taking both distinctiveness and the number of distinct designs into account. We hope our work could inspire future work to develop even more reasonable and hopefully universally acceptable metrics for diversity.

- Last but not least, while our work is based on simulation, we note that there is ongoing research on the realization of soft robotics in the physical world, using polymers with pneumatic chambers (Kriegman et al., 2020b; Legrand et al., 2023) or even self-replicating cells (Kriegman et al., 2020a; 2021) and continually narrowing the sim-to-real gap. We believe that with the collective efforts of material scientists, computer scientists, (bio)mechanical engineers, etc., soft robotics would see rapid advances and finds its way to everyday life in the near future.

## O  AN ANALYSIS OF COMPUTATIONAL EFFICIENCY

According to the data released on LLM Leaderboard (https://artificialanalysis.ai/leaderboards/models), for GPT-4o-mini, the median rate of output token generation is 99.8 tokens per second, and the latency (*i.e.,* time to first token) is reported as 0.5 seconds. Given that LASeR makes an average of 130 API calls per generation, with each call involving approximately 180 output tokens (here we assume the worst case where each newly generated robot design triggers DiRect), this results in an overhead of around 5 minutes per generation, or 5 hours in total.

The latency issue could be mitigated with locally deployed LLMs, which are less affected by network delays and request queuing. However, we believe it is more pertinent to compare the overall running time of different methods, with optimization efficiency taken into account. Specifically, by checking the log messages of our programs, we find that, for Carrier-v0, in order to reach the same level of fitness, LASeR requires **7 hours**, in contrast to the most competitive baseline, LLM-Tuner, which takes **15 hours**. For Pusher-v0, the difference is greater: LASeR requires **11 hours**, whereas LLM-Tuner takes **46 hours**. On Walker-v0, LASeR is even capable of reaching a fitness **unattainable by baselines**. Hence, the rapid convergence enabled by LLMs outweighs the additional computation overhead, rendering the latter perfectly worthwhile.

## P  PSEUDO CODE OF LASeR

---

**Algorithm 1:** LASeR: LLM-Aided Evolutionary Search for Robot Design Automation

---

**Input:** A task instance $T$, maximum number of evaluations $M$, population size $N$, survival rate $r$, maximum number of LLM interactions in each generation $L$, probability of similarity check $p$, similarity threshold $s$.

**Output:** $M$ robot designs together with their fitness scores.

Randomly initialize a population $\mathcal{P}$.     //Or initialize with zero-shot proposals from LLM

$\mathcal{S} \leftarrow \mathcal{P}$        //$\mathcal{S}$ keeps all the evaluated robot designs and their fitness scores

Warm start with several generations of a traditional EA.

**while** $|\mathcal{S}| < M$ **do**

  $attempts \leftarrow 0$          //Track the number of LLM interactions

  $\mathcal{P} \leftarrow$ the top $N \times r$ robot designs among $\mathcal{P}$; $survivors \leftarrow \mathcal{P}$  //Natural selection

  **while** $|\mathcal{P}| < N$ *and* $attempts < L$ **do**

    $target\_fitness \leftarrow$ max_fitness($\mathcal{S}$)$\times 1.2$ //Calculate target fitness as $1.2 \times$ current max

    $prompt \leftarrow \{$metadata of task $T$, $survivors$ in ascending order, $target\_fitness\}$

    $robot \leftarrow$ LLM($prompt$)      //Query the LLM for a new offspring solution

    $attempts \leftarrow attempts + 1$

    $u \sim \mathcal{U}(0,1)$

    **if** $u < p$ **then**

      //Enter similarity check with probability $p$

      Check the similarity rate $s'$ of $robot$ to existing robot designs.

      **if** $s' < s$ **then**

        $\mathcal{P} \leftarrow \mathcal{P} \bigcup \{robot\}$      //Pass the similarity check

      **else**

        $robot \leftarrow$ DiRect($robot$)   //Modify with Diversity Reflection Mechanism

        Check the similarity rate $s'$ of $robot$ to existing robot designs.

        **if** $s' < s$ **then**

          $\mathcal{P} \leftarrow \mathcal{P} \bigcup \{robot\}$   //Pass the similarity check after modification

  **if** $|\mathcal{P}| < N$ **then**

    //LLM has not generated enough robots

    Generate the remaining offspring with the traditional EA.

  **for** $robot$ *in* $\mathcal{P}$ **do**

    //Control optimization and fitness evaluation

    **if** $robot$ *not in* $survivors$ **then**

      Optimize a controller for $robot$. Evaluate the cumulative reward as its fitness $f$.

      $\mathcal{S} \leftarrow \mathcal{S} \bigcup \{(robot, f)\}$

**Return:** $\mathcal{S}$

---

