# OpenReview forum: "LASeR: Towards Diversified and Generalizable Robot Design with Large Language Models"
_ICLR.cc/2025/Conference — ICLR 2025 Poster_

### Official Review · Reviewer_drYB · 2024-11-01

**Soundness:** 3
**Presentation:** 3
**Contribution:** 2
**Rating:** 6
**Confidence:** 4

**Summary:**

This work proposes and evaluates an LLM-based evolutionary operator for robot design. The proposed method distinguishes itself from prior related work by incorporating an explicit mechanism for “reflection” and “inter-task knowledge transfer” where the former intends to balance exploitation and exploration and the latter intends to exploit existing robot design datasets whilst leveraging LLMs ability for inter-task reasoning.

**Strengths:**

Originality

This paper demonstrates originality by identifying the need for and explicitly developing a mechanism aimed at solving lack of diversity in LLM-aided robot design processes. The idea of using existing robot design datasets and inter-task reasoning to transfer or modify it also adds originality to this work.

Quality

The authors clearly set out their intended contributions and state their scientific hypotheses as well as the experiments they intend to test these hypotheses. The figures throughout the paper are high quality and aesthetically pleasant. The authors do a good job of covering a fairly broad portion of the related literature and providing motivations for their own work.

Clarity

The paper is well-written and easy to follow. The methods and results figures are easy to interpret. The main method figure is done particularly well and makes it easy to understand the relatively involved, multi-step process that is the proposed algorithm. The tables throughout the paper are also well-labeled without superfluous text.

Significance

Joint design and control of robotic systems is an important problem and provides a canonical example of a combinatorially explosive design space automated methods aspire to solve. The lack of diversity or tendency towards local optima in the morphological design process is a known limitation of evolutionary robotics broadly, so research and development in this area is crucial to advance the field.

**Weaknesses:**

Small design space

A 5x5, two dimensional design space makes it difficult to discover interesting structures or behaviors, rendering the implications for design (let alone robotics) somewhat unclear. The search space is much smaller than most work over the past three decades, which has been in 2D but at much higher resolution with hundreds of independent motors (https://www.roboticsproceedings.org/rss20/p100.html) or in 3D with hundreds or thousands of voxels (https://www.creativemachineslab.com/soft-robot-evolution.html). The paper would be much more compelling if LLMs were operating over a 3D space where variety in gait patterns more readily appear and the control complexity increases substantially.

Weak notions of diversity and lacking examples

The paper proposes the measure diversity in terms of the voxel space edit distance of robots and the number of distinct, high-performing robots. The latter is likely a poor measure of diversity as two robots can be highly similar while remaining distinct in terms of the precise voxel layout, and the former is difficult to interpret. Moreover, the paper provides no examples of the morphologies (and diversity) discovered by their algorithm. The overall diversity measure is the weighted average of these two metrics with weights of 1.0 and 0.1. There is no rationale for the selection of these weights outside of an anecdote that weighting the latter by 0.1 makes the two “roughly on the same scale and given equal importance”.

Diversity reflection incomplete ablation

Following up on the above point, the paper reports an ablation study to test the effectiveness of their diversity reflection mechanism; however, the ablation does not elucidate whether the LLM is actually providing intelligent mutations that encourage diversity. An additional ablation study should be run wherein random mutations are made to existing designs (in parallel to LLM guided mutations). This would help demonstrate whether the diversity reflection mechanism represents an intelligent operator.

Early convergence

Also related to diversity, the proposed algorithm appears to converge very early relative to some other baselines in most cases. This appears to be an indication that the algorithm may be stuck in a local optima, or is it closer to a global optima? If allowed to run for longer would the other methods arrive at a similarly high performing result? If it is indeed discovering something that is close to globally optimal then the fast convergence should indicate that this task (read design space) is too easy.

It also appears that when using the diversity reflection component the algorithm converges faster, which seems somewhat counter-intuitive as one would expect greater exploration to produce longer convergence times. The fact that it does not leads back to a prior question as to whether the LLM is truly modifying the design in intelligent ways that ultimately produce meaningful diversity in the population.

Marginal gains in performance

When all is said and done the proposed method produces marginal gains in performance relative to baselines. In the ablation study with and without diversity reflection performance also does not change substantially whereas the diversity does improve significantly. The diversity metric itself remains questionable (see above).

The knowledge transfer mechanism also does not appear to itself demonstrate meaningful improvements relative to the LASeR without knowledge transfer.

Reproducibility

The paper states that all experiments are conducted three times and the results are averaged. Since the performance gains are relatively small, additional trials are necessary to demonstrate statistical significance of the results.

Missing related work

This paper fails to discuss, compare and contrast their work with other similar methods that use LLMs to design robots, for example: https://link.springer.com/chapter/10.1007/978-981-99-3814-8_11

Choices about the control policy and its training also bears at the very least some discussion and comparison to other evolutionary robotics approaches that employ other methods, such as gradient based optimization (https://www.roboticsproceedings.org/rss20/p100.html), for the control problem.

**Questions:**

1. How do you justify the proposed measures of diversity and the weighted averaging used? Can you provide other measures? For example, the proportion of bodies made up of different voxel types?

2. Can you provide concrete examples of morphologies that emerged using your method compared with others? Is the diversity in these collections immediately observable just by looking at the bodies?

3. Can you run these experiments again several times over to provide more meaningful performance measures, confidence intervals and statistical hypothesis testing?

4. How do you explain the early convergence of your method when a primary claim relates to encouraging population level diversity and design exploration?

5. Can you perform additional ablation studies of the diversity reflection aspect? For example, randomly edit voxels and compare results to LLM-based editing? Can you catalog examples of LLM edits that encourage diversity through an evolutionary lineage?

6. Can you run experiments with a larger design space? Perhaps 9x9?

---

> ### Author Response · Authors · 2024-11-26
> **Response to Reviewer drYB (1/5)**
>
> Dear reviewer,
>
> Thank you for your constructive feedback. The following is our detailed response to the questions you raised. Please let us know if you have any trouble accessing our revised paper.
>
> >**Weakness 1 (Small design space)**
>
> >**Question 6**: Can you run experiments with a larger design space? Perhaps 9x9?
>
> **Response**:
> Thank you for the question. Although our approach is conceptually adaptable to larger design spaces, such as those consisting of hundreds or thousands of functional units, several challenges would arise when scaling to such environments. For one, the increased degrees of freedom would result in **a surge in control complexity**. To address this, existing works typically employ either periodic **open-loop actuation** patterns for locomotion on relatively simple terrains (Cheney et al., 2014), or **differentiable simulations** to facilitate more sample-efficient control learning (Strgar et al., 2024; Cochevelou et al., 2023). The former is **limited in its applicability to more complex tasks**, such as those involving varying terrains or object manipulation, while the latter introduces **considerable computational and memory overhead** (Cochevelou et al., 2023).
>
> These challenges are likely why most existing voxel-based soft robot (VSR) studies focus on smaller design spaces, such as 5x5 configurations (Song et al., 2024; Saito and Oka, 2024; Dong et al., 2024; Wang et al., 2023; Bhatia et al., 2021). On one hand, a 5x5 body size with five material types yields a design space of $2.98\times10^{17}$ possible solutions, which is **sufficiently expressive for generating complex and diverse morphological structures**. On the other hand, the infinite degrees of freedom inherent in soft materials make it **nearly impossible to substantially scale up robot sizes without sacrificing the ability to evaluate VSRs in more demanding task settings**. Nonetheless, we believe this dilemma can be mitigated with advances in more efficient simulation engines and control algorithms.
>
> For now, we have tried a 10x10 Walker environment (Appendix I in revised paper), where **LASeR still holds advantage**. We are excited to evaluate our approach in even larger design spaces in our future work. We note that the current results are averaged across two independent runs. We planned to conduct three runs but there remains one of them unfinished. We would post the complete results once they are available. For your reference, we notice that our experiments on 10x10 Walker-v0 take on average **1.5 times longer** than the 5x5 case.

---

> > ### Comment · Reviewer_drYB · 2024-11-26
> >
> > There appears to be little to no appreciable difference between LASeR and the provided baseline in the larger design space, if there is the effect size is once again very very small. That said, it is good to know that the method scales to larger design spaces. It will be interesting to see how the additional trials and statistics turn out. Thank you for adding this to the paper.

---

> ### Author Response · Authors · 2024-11-26
> **Response to Reviewer drYB (2/5)**
>
> >**Weakness 2 (Weak notion of diversity and lacking examples)**
>
> >**Question 1**: How do you justify the proposed measures of diversity and the weighted averaging used? Can you provide other measures? For example, the proportion of bodies made up of different voxel types?
>
> >**Question 2**: Can you provide concrete examples of morphologies that emerged using your method compared with others? Is the diversity in these collections immediately observable just by looking at the bodies?
>
> **Response**:
> Thank you for the question, and we appreciate this opportunity to clarify our approach towards diversity measurement. First of all, we acknowledge that the number of distinct, high-performing robots is not a comprehensive measure of diversity, as it does not account for the extent of distinctiveness of these robots. Rather, this measure was designed as a ***correction* to prevalent diversity metrics**. Previous studies have predominantly employed two methods for quantifying diversity: (1) averaged measures of distinctiveness within a group of robots, such as per-voxel entropy (Song et al., 2024) or pairwise edit distance (Saito and Oka, 2024); (2) manual categorization of robot designs into distinct classes, followed by the computation of the Simpson index, which is analogous to an entropy of class distribution (Medvet et al., 2021). However, **the latter method becomes impractical** when dealing with more abstract morphologies that lack clear subpopulations. The former approach presents a ***paradox*** (as illustrated in Appendix L in revised paper): adding a new robot design to an existing collection can reduce diversity, even if the new design is distinct, provided that it falls within the existing distribution of this collection. This is **counter-intuitive** because the addition of an alternative should increase, rather than decrease, overall diversity.
>
> To address the above issue, we incorporate the number of distinct robot designs into our measurement as a correction. Thus, our two measures -- **distinctiveness** (through edit distance) and **the number of distinct designs** -- complement each other, providing a **more comprehensive and reasonable** characterization of diversity. **To clarify, the edit distance measure amounts to counting the number of different voxels between a pair of robot designs, and this should be equivalent to “the proportion of bodies made up of different voxel types” that you mentioned.**
>
> While our approach takes both distinctiveness and the number of robots into account, we acknowledge that the weights assigned to these quantities are somewhat expedient and primarily intended to bring them onto the same scale. Specifically, we selected weights of 1.0 and 0.1 based on preliminary experiments, where we found that the number of distinct high-performing designs obtained in a single run of experiment ranged **from several dozen to around two hundred**, while the edit distance is defined to range **between 0 and 25**. Given the lack of universally accepted metrics for morphological diversity, we hope our approach will inspire future work to propose more reasonable and comprehensive measurements. In response to your suggestion, we have also visualized the robot designs evolved by different methods to enable a more straightforward and intuitive comparison of diversity (Appendix K of revised paper).

---

> > ### Comment · Reviewer_drYB · 2024-11-26
> >
> > Acknowledging that diversity is difficult to quantify is helpful, thank you for adding it to the paper. However, the arbitrary weighting coefficients renders the employed metric difficult to interpret. I'd suggest reporting diversity in terms of edit distance, # of distinct high performers, and the weighted average separately so the reader can compare different methods. Demonstrating that LASeR outperforms on all three would be convincing, and where LASeR underperforms there may be insight into the limitations of the existing metrics.

---

> ### Author Response · Authors · 2024-11-26
> **Response to Reviewer drYB (3/5)**
>
> >**Weakness 3 (Diversity reflection incomplete ablation)**
>
> >**Question 5**: Can you perform additional ablation studies of the diversity reflection aspect? For example, randomly edit voxels and compare results to LLM-based editing? Can you catalog examples of LLM edits that encourage diversity through an evolutionary lineage?
>
> **Response**:
> We appreciate your comment, which is really thought-provoking. We agree that comparing our diversity reflection mechanism with random editing would provide a more compelling evaluation. In response, we re-implemented our experiments with DiRect replaced by random voxel mutations. We found that for DiRect, the fitnesses of robot designs before and after DiRect modification show **no significant difference** (**$p$=0.19**). In contrast, the fitnesses of randomly mutated robot designs are **significantly lower** than their pre-editing counterparts (**$p$<0.001**). The evolution with random editing also suffers from **reduced optimization efficiency** (Appendix F in revised paper), as the “uninformed” exploratory behavior often disrupts essential functional structures. LLM-aided diversity reflection, on the other hand, builds on successful designs discovered along the evolutionary trajectory, and hence **promotes exploration without compromising optimization efficiency**. We believe this also addresses the question you raised in the second paragraph of Weakness 4 (early convergence). Specifically, the “***informed***” rather than random mutations of LLM is the key reason for the faster convergence, as it promotes more thorough exploration of the design space while keeping the functionality of robot designs largely intact.
>
> To further address your concerns and demonstrate that the LLM is indeed providing intelligent mutations, we present **some examples of diversity reflection** throughout the evolutionary process (Appendix F in revised paper). These examples include both the pre- and post-editing morphologies, along with explanations provided by the LLM for its decision making. We find that the LLM is able to **identify critical substructures** within robot designs and **modifies only the voxel placements that do not affect functionality**, yet promote diversity. We believe these results provide strong evidence that the diversity reflection mechanism is reliably functioning as an ***intelligent* mutation operator**.
> _____
> >**Weakness 4 (Early convergence)**
>
> >**Question 4**: How do you explain the early convergence of your method when a primary claim relates to encouraging population level diversity and design exploration?
>
> **Response**:
> Thank you for your feedback. To address your concern about early convergence, we re-implemented LLM-Tuner, the most competitive baseline, for 2000 robot evaluations -- double the number used in our original experiments -- across three independent repetitions. The averaged results are presented in Appendix D of revised paper. Notably, **LLM-Tuner does not end up with higher fitness levels** than those achieved by LASeR, largely confirming that our algorithm **has not been stuck in local optima**. Meanwhile, the evidently slower convergence of LLM-Tuner (especially pronounced in Walker-v0 and Pusher-v0) indicates that our fast convergence is more likely due to the **effectiveness afforded by LLM-aided evolution and diversity reflection**, rather than an artifact of task difficulty.

---

> > ### Comment · Reviewer_drYB · 2024-11-26
> >
> > Please clarify how random voxel editing was implemented.
> >
> > How did the number of edited voxels compare to the number edited by the LLM operator?
> >
> > How does Fig 13 represent superiority of an LLM operator over random mutations?
> >
> > The performance difference seems negligible, and with a single trial and large differences in initialization, it is not so convincing.
> >
> > Also, if the random editing made designs worse then why does performance of the dashed red line seem to improve over evolutionary time?

---

> ### Author Response · Authors · 2024-11-26
> **Response to Reviewer drYB (4/5)**
>
> >**Weakness 5 (Marginal gains in performance)**
>
> >**Weakness 6 (Reproducibility)**
>
> >**Question 3**: Can you run these experiments again several times over to provide more meaningful performance measures, confidence intervals and statistical hypothesis testing?
>
> **Response**:
> We appreciate your valuable suggestion, and performed two more sets of repeated experiments. However, due to limited computation resources, we were only able to run these experiments on LASeR and LLM-Tuner (the most competitive baseline). We plan to continue with the remaining baselines and include the complete results once they are available.
>
> As shown in Appendix H of revised paper, with a total of 5 repeated experiments, the advantage of LASeR over LLM-Tuner **remains evident**, in terms of both optimization efficiency and diversity. Moreover, we would like to clarify that we have chosen a **sufficiently large number of robot evaluations** to hopefully allow all baseline algorithms to converge, enabling fair comparison. This explains why some baselines have ended up with rather similar evolutionary outcomes to LASeR. However, in the context of robot design automation, besides the final fitness level, **the speed at which high-performing designs are approached is an equally important aspect for evaluating design algorithms**. This is due to the heavy computational burden involved in training control policies and the manufacturing costs of physical robots when deployed in real-world applications. In this regard, LASeR achieves **considerable performance gains**. This is particularly notable in Carrier-v0 and Pusher-v0, where LASeR requires **2x** and **3x** fewer evaluations, respectively, to reach the same level of fitness as the best-performing baseline. The generally non-overlapping confidence intervals of fitness curves in Appendix H of revised paper further support the **statistical significance of this superiority**.
>
> However, we notice that both LASeR and LLM-Tuner exhibit relatively high variability in their diversity outcomes, suggesting that even more repetitions would be needed to establish statistical significance. Therefore, we will continue with additional repeated experiments.

---

> > ### Comment · Reviewer_drYB · 2024-11-26
> >
> > >...the advantage of LASeR over LLM-Tuner remains evident, in terms of both optimization efficiency and diversity.
> >
> > >The generally non-overlapping confidence intervals of fitness curves in Appendix H of revised paper further support the statistical significance of this superiority.
> >
> > The advantage is not evident and the results do not support statistical significance.
> >
> > It would be helpful if the authors can plot a 95%+ confidence interval instead of a standard deviation, conduct a statistical test, and report the p-val.
> >
> > >However, we notice that both LASeR and LLM-Tuner exhibit relatively high variability in their diversity outcomes, suggesting that even more repetitions would be needed to establish statistical significance. Therefore, we will continue with additional repeated experiments.
> >
> > This is the opposite of what you just said!

---

> ### Author Response · Authors · 2024-11-26
> **Response to Reviewer drYB (5/5)**
>
> >**Weakness 7 (Missing related work)**
>
> **Response**:
> Thank you for pointing out this issue. While we have discussed this particular paper (Lehman et al., 2023) in our related work section, we chose not to compare against it because it was originally proposed for **designing Sodaracers via code optimization**, which necessitates a **program interface** that maps between robot designs and Python codes. This mapping becomes difficult to define in the context of voxel-based soft robots. To further demonstrate the effectiveness of our approach, we conducted additional comparisons with another state-of-the-art LLM-based evolutionary algorithm, OPRO (Yang et al., 2024), which operates in the original solution space as we do. The results, presented in Appendix G, still show **a clear advantage of LASeR**.
>
> Regarding the choice of control policy and its training, we have adhered to the **standard practices** used in previous EvoGym-based studies, specifically the proximal policy optimization (PPO) algorithm with MLPs as control networks. While gradient-based optimization would be an interesting direction to explore, it is unfortunately inapplicable to EvoGym as it **does not support differentiable simulation**. We have clarified our rationale for choosing this control algorithm in our revised paper, and will explore a broader range of control approaches in future work, examining their potential impact on our proposed evolutionary framework.
> _____
> Thank you again for your time and effort spent reviewing this paper! Your suggestions have been really conducive to our revision. Please let us know if your questions and concerns have been fully addressed.
>
> **References**:
>
> [1] Bhatia, Jagdeep, et al. "Evolution gym: A large-scale benchmark for evolving soft robots." Advances in Neural Information Processing Systems 34 (2021): 2201-2214.
>
> [2] Cheney, Nick, et al. "Unshackling evolution: evolving soft robots with multiple materials and a powerful generative encoding." ACM SIGEVOlution 7.1 (2014): 11-23.
>
> [3] Cochevelou, François, David Bonner, and Martin-Pierre Schmidt. "Differentiable soft-robot generation." Proceedings of the Genetic and Evolutionary Computation Conference. 2023.
>
> [4] Dong, Heng, Junyu Zhang, and Chongjie Zhang. "Leveraging Hyperbolic Embeddings for Coarse-to-Fine Robot Design." The Twelfth International Conference on Learning Representations. 2024.
>
> [5] Lehman, Joel, et al. "Evolution through large models." In Handbook of Evolutionary Machine Learning, pp. 331–366. Springer, 2023.
>
> [6] Medvet, Eric, et al. "Biodiversity in evolved voxel-based soft robots." Proceedings of the Genetic and Evolutionary Computation Conference. 2021.
>
> [7] Strgar, Luke, et al. "Evolution and learning in differentiable robots." arXiv preprint arXiv:2405.14712 (2024).
>
> [8] Saito, Takumi, and Mizuki Oka. "Effective Design and Interpretation in Voxel-Based Soft Robotics: A Part Assembly Approach with Bayesian Optimization." Artificial Life Conference Proceedings 36. Vol. 2024. No. 1. One Rogers Street, Cambridge, MA 02142-1209, USA journals-info@ mit. edu: MIT Press, 2024.
>
> [9] Song, Junru, et al. "MorphVAE: Advancing Morphological Design of Voxel-Based Soft Robots with Variational Autoencoders." Proceedings of the AAAI Conference on Artificial Intelligence. Vol. 38. No. 9. 2024.
>
> [10] Wang, Yuxing, et al. "PreCo: Enhancing Generalization in Co-Design of Modular Soft Robots via Brain-Body Pre-Training." Conference on Robot Learning. PMLR, 2023.
>
> [11] Yang, Chengrun, et al. “Large Language Models as Optimizers.” The Twelfth International Conference on Learning Representations. 2024.

---

> > ### Comment · Reviewer_drYB · 2024-11-26
> >
> > The robot design problem reads as an afterthought and this field of work and its rich history are glossed over, pushed to the very end of the paper and the appendix.
> >
> > Why is robot design important?
> >
> > Why should we care about this problem?
> >
> > You call your agents "robots" but failed to explain how they can transfer from 2D simulation to reality? Has this been done before? How?
> >
> > Are there any implications of this work for the future of real robots?

---

> > > ### Author Response · Authors · 2024-11-28
> > > **Further Response to Reviewer drYB by Authors (5/5)**
> > >
> > > Thank you for the question. We have no intention to gloss over previous works on robot design, as they unarguably inspired generations of researchers (including us) and led to today’s state of the art. In our related work section, we have discussed the history of robot design automation, from Sims’ pioneering work that dates back to 1990s all the way to more recent works that resort to combinations of evolutionary algorithms and deep probabilistic generative models, and to today’s work that leverages LLMs. Now, to pay due respect to all these works and their contributions, we have moved Related Work to Section 2, right after Introduction.
> > >
> > > **Robot design automation** is an established research problem in Robotics that aims to automatically evolve robot morphology according to specific task objectives with minimal human intervention. We believe the primary reason why this is considered an important research field is the recognized significance of morphology to intelligent behavior (known as ***morphological intelligence***) (Ghazi-Zahedi, 2019; Gupta et al., 2021). We share common view with you that robot design automation should eventually be applied to physical robots to truly benefit humankind, which is also our long-term vision. However, due to the prohibitive labor and manufacturing costs involved in deploying physical robots, it has become nearly a consensus that **design algorithms should be developed and prototyped in simulation**. To this end, numerous unselfish researchers developed simulation environments that are efficient and easy to use, and meanwhile emulate real-world physics with high fidelity. The development of these environments usually involves considerable efforts and cross-disciplinary knowledge (such as materials science and mechanics), but once developed, would become reliable platforms that subsequent works (involving both robot design and control) could build upon. In this work, we employed **Evolution Gym** (Bhatia et al., 2021), a latest, dedicated simulation environment developed by MIT specifically for voxel-based soft robots (VSRs). Specifically, EvoGym utilizes a classic **mass-spring system with cross-braced traction** to simulate elastic building blocks. Many other considerations, such as bounding box trees for collision detection and computation of penalty-based contact forces and frictional forces, are taken to ensure the simulation fidelity. We have chosen EvoGym also due to its **popularity in past literature**, which proves its reliability. There are numerous simulation environments other than EvoGym that are also available, including **2D-VSR-Sim** (Medvet et al., 2020) and **diffTaichi** (Hu et al., 2019), serving as the basis of hundreds of works each year and greatly contributing to the advances of robot design automation.
> > >
> > > At the meantime, there is also ongoing research on the realization of soft robotics in the physical world, using polymers with pneumatic chambers (Kriegman et al., 2020 (b); Legrand et al., 2023) or even self-replicating cells (Kriegman et al., 2020 (a) and 2021) and **continually narrowing the sim-to-real gap**. These studies also revealed promising avenues through which our approach could be applied in real world. We have cited these papers with a brief discussion in Section 2. We believe that with the collective efforts of material scientists, computer scientists, (bio)mechanical engineers, etc., soft robotics would see rapid advances and finds its way to everyday life in the near future. One of the major implications of our particular work to robot design is that it **reveals a remarkable potential of Large Language Models to design morphology**, with the aid of a meticulously designed reflection mechanism. This, together with rapidly progressing foundation models (FMs) and FM-aided control strategies, points to a promising prospect where intelligent agents are capable of both designing and controlling their own embodiments.

---

> > > > ### Author Response · Authors · 2024-11-28
> > > > **References**
> > > >
> > > > **References:**
> > > >
> > > > [1] Bhatia, Jagdeep, et al. "Evolution gym: A large-scale benchmark for evolving soft robots." Advances in Neural Information Processing Systems 34 (2021): 2201-2214.
> > > >
> > > > [2] Ghazi-Zahedi, Keyan. "Morphological Intelligence." Cham: Springer (2019).
> > > >
> > > > [3] Gupta, Agrim, et al. "Embodied intelligence via learning and evolution." Nature communications 12.1 (2021): 5721.
> > > >
> > > > [4] Hu, Yuanming, et al. "Difftaichi: Differentiable programming for physical simulation." arXiv preprint arXiv:1910.00935 (2019).
> > > >
> > > > [5] Kriegman, Sam, et al. "A scalable pipeline for designing reconfigurable organisms." Proceedings of the National Academy of Sciences 117.4 (2020,a): 1853-1859.
> > > >
> > > > [6] Kriegman, Sam, et al. "Scalable sim-to-real transfer of soft robot designs." 2020 3rd IEEE international conference on soft robotics (RoboSoft). IEEE, 2020 (b).
> > > >
> > > > [7] Kriegman, Sam, et al. "Kinematic self-replication in reconfigurable organisms." Proceedings of the National Academy of Sciences 118.49 (2021): e2112672118.
> > > >
> > > > [8] Legrand, Julie, et al. "Reconfigurable, multi-material, voxel-based soft robots." IEEE Robotics and Automation Letters 8.3 (2023): 1255-1262.
> > > >
> > > > [9] Medvet, Eric, et al. "2D-VSR-Sim: A simulation tool for the optimization of 2-D voxel-based soft robots." SoftwareX 12 (2020): 100573.
> > > >
> > > > [10] Song, Junru, et al. "MorphVAE: Advancing Morphological Design of Voxel-Based Soft Robots with Variational Autoencoders." Proceedings of the AAAI Conference on Artificial Intelligence. Vol. 38. No. 9. 2024.

---

> > > > > ### Comment · Reviewer_drYB · 2024-11-28
> > > > >
> > > > > I appreciate the author’s effort and believe that with the promised repeated trials this paper should be published. I am raising my score to reflect this.

---

> > > > > > ### Author Response · Authors · 2024-11-29
> > > > > > **Response by Authors**
> > > > > >
> > > > > > We greatly appreciate your recognition of our work, which is very encouraging. Thank you again for your time and effort spent reviewing this paper.

---

> ### Author Response · Authors · 2024-11-28
> **Further Response to Reviewer drYB by Authors (1/5)**
>
> Dear reviewer,
> We very much appreciate your active participation in the discussion, and the questions you raised are very instructive. Our responses are as follows.
>
> We appreciate this opportunity to clarify that the absolute difference in fitness largely depends on **how the reward function of the task instance is defined**, as well as **where the true difficulty of a certain task lies in**. From our observation, locomotion tasks in EvoGym, regardless of the scale of design spaces, generally exhibit these characteristics: (a) Even randomly initialized populations could achieve **decent fitness levels**, due to the powerful RL algorithms and relatively low requirement for a robot design to pick up at least some form of walking gaits (it remains to be verified whether this will hold in even larger design spaces, such as 100x100); (b) However, most of the challenge lies in, once a design algorithm reaches a certain level of fitness, will it be able to **break through the bottleneck** and achieve further performance gains (This bottleneck effect is also demonstrated in Song et al. (2024)). The latter aspect demands high proficiency in a search algorithm to conduct rather **fine-grained optimization w.r.t. voxel placements**. This explains why on Walker-v0 (both 5x5 in Figure 13,15,16 and 10x10 in Figure 17), different algorithms typically reach very similar fitness levels in early stages of evolution. But **it is instead the nuanced differences thereafter that truly reflect optimization efficiency**. On the contrary, we note that object manipulation tasks (such as Pusher-v0, Carrier-v0 and Catcher-v0) have the opposite nature: they typically require **more coarse-grained** (but not necessarily easy-to-find) functional substructures to have optimal task performance, and hence **the early-stage convergence speed deserves more attention**. To further establish the statistical significance of our approach and render our evaluations more convincing, we will conduct more repeated experiments and post the results once they are available.

---

> ### Author Response · Authors · 2024-11-28
> **Further Response to Reviewer drYB by Authors (2/5)**
>
> Thank you for the suggestion. We now provide both the **separate measures** (Table 6 and 7 in our latest revised paper) and their **weighted average** (Table 8) in our paper. These results also include two additional SOTA baselines and Catcher-v0 (a hard task). Our finding suggests that **LASeR has more of a advantage in discovering distinct high-performers** than achieving high averaged edit distance. We cannot actually state which approach is more favorable, as both benefit diversity in their own way. However, combining the results of optimization efficiency (i.e. the fitness curves), it is clear that **LASeR better balances exploration with exploitation**.
>
> Meanwhile, as we pointed out in Appendix L, we believe that the results in Table 7 are somewhat misleading, because edit distance in itself does not suffice as a valid diversity metric. Even if a group of robot designs has another group as its subset, the former might still have a lower edit distance (even much lower, due to the paradox in Figure 20). The comparative results in Table 7 might not be in our favor, but we choose to display them to **reveal an open problem regarding diversity measurement** and hopefully inspire future work to investigate further.
>
> We additionally come up with yet another diversity metric -- **the total number of different voxels between all pairs of high-performing robots** (i.e. edit distance without being averaged). We believe this metric more naturally aggregates distinctiveness and number of distinct high performers **without needing pre-specified weighting coefficients**, thus benefiting from **better interpretability**. The results are reported in **Table 9**, which show that LASeR ranks first in Walker-v0 and second in the remaining tasks. The major competitor here is MorphVAE, which achieves an average rank of 2.5 across four tasks. LASeR, on the other hand, achieves 1.75. This means that, according to this newly proposed metric, **LASeR still achieves the highest overall diversity**.

---

> ### Author Response · Authors · 2024-11-28
> **Further Response to Reviewer drYB by Authors (3/5)**
>
> We implemented random editing by substituting the diversity reflection (DiRect) mechanism with random voxel mutation, which is supported by a built-in function of EvoGym. Specifically, we found that the number of voxels edited by DiRect in each design is about **2.61** on average. Thus, for random mutation, we set the mutation rate to be 0.1, i.e. each voxel will, with a probability of **0.1**, be randomly replaced by a different material. Given that a robot design consists of 25 voxels, this results in **2.5** voxels being edited on average, which we believe is reasonably close to DiRect editing.
>
> In Figure 13, we observe that both LASeR w/ DiRect and LASeR w/ random editing can reach a comparable level of fitness during early stage of evolution. This once again reflects the bottleneck effect explained in the response to Question 1. However, LASeR w/ DiRect demonstrated **continuous improvements** after this, whereas LASeR w/ random editing **stagnated a lot**, represented by the long segments of flat fitness curve. These observations suggest that the proposed diversity reflection mechanism better facilitates fine-grained optimization of voxel combinations due to its “***informed***” nature. The “negligible” performance difference is primarily due to how the reward function is defined, as well as where the true difficulty of a task lies in (as explained in the response to Question 1). In Walker-v0, the reward function is defined as the distance travelled in given time. However, we notice that even randomly generated robot morphology can achieve decent travelling distance with optimized control policies. It is, instead, **the further improvements beyond 10.6 that reflect how a morphology is truly adapted to locomotion**. But we do agree that more repeated experiments are needed to verify statistical significance of our superiority. We will post the results once these experiments are finished.
>
> The dashed red line improves because here random editing is **only replacing the DiRect mechanism** rather than the whole search algorithm. That is, we are still using LLM as the search operator that generates offspring solutions and using natural selection to keep top-ranking solutions among them. Random editing is playing the part of DiRect to introduce variability into offspring solutions when they overly resemble evaluated ones, except that it relies on random mutation rather than a reflection mechanism. We implemented this set of experiments mainly to show that **the proposed diversity reflection mechanism is indeed introducing variability in a more *intelligent* way**.

---

> ### Author Response · Authors · 2024-11-28
> **Further Response to Reviewer drYB by Authors (4/5)**
>
> Thank you for the constructive feedback. Following your advice, we plot 95% confidence intervals (i.e. mean±1.96std) rather than one standard deviation in Figure 16. We also conduct hypothesis testing (specifically two-tailed $t$-test) to prove statistical significance. Since in this work we have chose **a sufficiently large number of robot evaluations** (i.e. 1000) to give ample opportunity to all algorithms to converge, it becomes more relevant to **compare the convergence speed** rather than entire fitness curves. To this end, we first average the eventual fitness values obtained by all repeated experiments (denoted as $f$) within a given task, and then record the number of evaluations that each experiment took to reach this average fitness (denoted as $n$). For those that did not reach $f$, $n$ is simply recorded as 1000. We then conduct a two-tailed $t$-test to compare the $n$’s of different algorithms. For Carrier-v0, $f$ is 10.69, and $n$ is on average **719.2** and **979** for LASeR and LLM-Tuner, respectively (**$p$=0.029**). For Pusher-v0, $f$ is 12.95, and $n$ is on average **528** and **888.6** with **$p$=0.054**. For Walker-v0, since none of the experiments of LLM-Tuner reach $f$=10.65, we instead compare the eventual fitness values achieved by LASeR and LLM-Tuner, which are on average **10.67** and **10.63**, with **$p$<0.001**. Since we are comparing against the best-performing baseline (which itself is **SOTA without much room left for improvement**), we believe the above analysis confirms the significant advantage of LASeR **in terms of optimization efficiency**. With regard to diversity, we notice that both LASeR and LLM-Tuner (as well as many other baselines) exhibit high variability in their diversity outcomes. So more than 5 repeated experiments would be needed to prove statistical significance. However, given that **LASeR nearly consistently achieves top-ranking diversity across different tasks** (as seen in Table 1 and 9), we believe this largely verifies the robustness of its advantage. Nevertheless, we will continue with additional repeated trials.
>
> >“However, we notice that both LASeR and LLM-Tuner exhibit relatively high variability in their diversity outcomes, suggesting that even more repetitions would be needed to establish statistical significance. Therefore, we will continue with additional repeated experiments.”
>
> Our above argument is with respect to diversity, rather than optimization efficiency. Sorry for causing the misunderstanding.

---

### Official Review · Reviewer_5ABx · 2024-11-03

**Soundness:** 3
**Presentation:** 4
**Contribution:** 3
**Rating:** 8
**Confidence:** 4

**Summary:**

The paper investigates the use of large language models in designing and evolving robots. To this end the paper uses an LLM to reflect and propose novel 'soft' robot designs in simulation in an evolutionary design loop. The paper compares its proposed LLM-evolution loop against different baselines and presents in-depth ablations of different effects LLM parameters have on the design loop.

**Strengths:**

- To the best of my knowledge the proposed framework is novel, the usage of LLMs in the problem of robot designs and their evolutions is under-researched
- The conclusions the paper makes, and its application are relevant to the robot learning community
- The paper compares its proposed approach versus several baselines
- The performed ablation studies are very interesting and insightful. I appreciate them.

**Weaknesses:**

Weaknesses:
- The environments in which the method is tested are relatively simple. However, I appreciate the hardness of the overall problem; designing and evolving robot hardware is not easy.
- A critical remark is that while the mean shows (in plots and tables) that the proposed method works, I think it is likely not statistically significant due to the standard deviation and the closeness of the final means.
- I think the paper could overall more critically discuss its limitations and open problems.
- The paper should probably cite and discuss this preliminary work discussing the potential of LLMs for the robot design process: Stella, Francesco, Cosimo Della Santina, and Josie Hughes. "How can LLMs transform the robotic design process?." Nature machine intelligence 5, no. 6 (2023): 561-564.

**Questions:**

I have no questions, overall I think the paper is in a good state and interesting to the ML/robotics community.

---

> ### Author Response · Authors · 2024-11-26
> **Response to Reviewer 5ABx (1/2)**
>
> Dear reviewer,
>
> We sincerely appreciate your recognition of the significance of our research problem and the contributions of our work. Please let us know if you have any trouble accessing our revised paper.
>
> **Response to Weakness 1**:
>
> Thank you for the comment, which makes perfect sense. In this work, we chose the relatively simple experimental setup, namely a 5x5 body size with 5 different material types, mainly to facilitate **a easier comparison** with the majority of related studies in this field, as it is commonly adopted in previous Voxel-based Soft Robot (VSR) research (Song et al., 2024; Saito and Oka, 2024; Dong et al., 2024; Wang et al., 2023). We believe the primary reason why they stick with this configuration is because it results in a combinatorially vast design space with approximately $2.98\times10^{17}$ possible solutions, already representing a challenging optimization problem. Meanwhile, it is also **expressive enough to generate complex and diverse morphological structures**. That being said, we find that our approach is readily adaptable to larger design spaces. Specifically, we performed additional evaluations on 10x10 Walker-v0, and observed that LASeR still **holds notable advantage** (Appendix I of revised paper). We look forward to seeing our approach applied to broader scenarios in future work. Furthermore, we conducted additional experiments on Catcher-v0 (Appendix E of revised paper), one of the most challenging tasks in EvoGym, and observed that our approach **continues to demonstrate significant advantages**. These promising results reveal the potential of our approach to scale to even larger and more complex design problems.

---

> ### Author Response · Authors · 2024-11-26
> **Response to Reviewer 5ABx (2/2)**
>
> **Response to Weakness 2**:
>
> To further demonstrate the effectiveness of our approach, we performed two additional sets of repeated experiments. However, we apologize for only being able to conduct these experiments on LASeR and LLM-Tuner (the most competitive baseline), due to limited computational resources. We plan to continue with the remaining baselines and include the complete results once they are available.
>
> Both the averaged results and standard deviations of the aforementioned repeated experiments are presented in Appendix H. With a total of five repeated runs, **the superiority of LASeR remains obvious**. The generally non-overlapping confidence intervals of fitness curves clearly indicate the **robust superiority** of our method in terms of optimization efficiency. In response to your concern regarding the closeness of final means, we would like to clarify that, for the sake of fair comparison, we **deliberately chose a sufficiently large number of robot evaluations** to hopefully let all baseline algorithms to converge. Despite this, our approach still achieves a fitness level that is **unattainable** by baseline methods in Walker-v0, and requires **2x** and **3x** fewer evaluations to reach optimal designs in Carrier-v0 and Pusher-v0, compared with the most competitive baseline.
>
> However, we did notice that both LASeR and LLM-Tuner exhibit relatively high variability in their diversity outcomes, and hence an even larger sample size would be needed to prove statistical significance. To this end, we will conduct additional repeated experiments. Thank you very much for your constructive feedback.
> _____
> **Response to Weakness 3**:
>
> We appreciate your value suggestion. In response, we have expanded the discussion of our limitations and outlined several open problems for future research in Appendix P.
> _____
> **Response to Weakness 4**:
>
> Thank you for highlighting this relevant article which presents an intriguing prospect of robotic design processes through interactive human-AI collaboration. We are pleased to be among the first to explore the potential of LLMs in automatically generating robot shapes, which we hope will contribute to the democratization of domain knowledge and enable non-specialists to develop effective robotic systems. We have cited this article in the revised version of our paper.
> _____
> We hope that we have fully addressed all your concerns. Once again, we sincerely thank you for your insightful feedback, which is greatly conducive to our revision.
>
> **References**:
>
> [1] Dong, Heng, Junyu Zhang, and Chongjie Zhang. "Leveraging Hyperbolic Embeddings for Coarse-to-Fine Robot Design." The Twelfth International Conference on Learning Representations. 2024.
>
> [2] Saito, Takumi, and Mizuki Oka. "Effective Design and Interpretation in Voxel-Based Soft Robotics: A Part Assembly Approach with Bayesian Optimization." Artificial Life Conference Proceedings 36. Vol. 2024. No. 1. One Rogers Street, Cambridge, MA 02142-1209, USA journals-info@ mit. edu: MIT Press, 2024.
>
> [3] Song, Junru, et al. "MorphVAE: Advancing Morphological Design of Voxel-Based Soft Robots with Variational Autoencoders." Proceedings of the AAAI Conference on Artificial Intelligence. Vol. 38. No. 9. 2024.
>
> [4] Wang, Yuxing, et al. "PreCo: Enhancing Generalization in Co-Design of Modular Soft Robots via Brain-Body Pre-Training." Conference on Robot Learning. PMLR, 2023.

---

> ### Author Response · Authors · 2024-11-29
> **We look forward to your feedback!**
>
> Dear Reviewer 5ABx,
>
> We deeply appreciate your favorable comments of our work, which have been very encouraging. We have revised our paper according to your thoughtful suggestions, and the modifications are listed as follows.
>
> - Included evaluations on a larger design space in Appendix J, as well as on a more complex task in Appendix E;
>
> - Included results from additional repeated experiments of LASeR and LLM-Tuner (the most competitive baseline) in Appendix H. These results are accompanied by significance tests to demonstrate the robust superiority of our approach;
>
> - Expanded our discussion of limitations and several open problems for future research in Appendix P;
> - Cited the highly relevant and inspiring article in Related Work (Section 2.2).
>
> We hope that with our responses and revisions, we have fully addressed all your questions and concerns. We also eagerly look forward to any further questions or comments you may have. Once again, we deeply appreciate the time and effort you have dedicated into reviewing our work. Happy Thanksgiving!

---

> ### Comment · Reviewer_5ABx · 2024-12-01
>
> I thank the authors for their responses to my raised questions and concerns.
> Overall, after the extensive discussion here I am still convinced that this paper is a valuable contribution and presenting an interesting first approach for using LLMs in a more interesting way in the robot design space and I remain positive on the recommendation to accept the paper.

---

> > ### Author Response · Authors · 2024-12-02
> >
> > We deeply appreciate your recognition, which is very encouraging. Thank you again for your time and effort!

---

### Official Review · Reviewer_D75R · 2024-11-04

**Soundness:** 3
**Presentation:** 3
**Contribution:** 2
**Rating:** 5
**Confidence:** 5

**Summary:**

In this paper, the authors introduce LASER (Large Language Model-Aided Evolutionary Search for Robot Design Automation), a novel approach that leverages Large Language Models (LLMs) to enhance the efficiency and diversity of robot design automation. In  LASER, LLM is integrated into the bi-level optimization framework, and the LLM is prompted to be the mutation operator for generating new robot morphologies.

**Strengths:**

1. The authors proposed a reflection mechanism for automated robot design, DiRect, to encourage more knowledgeable exploratory behaviors from LLMs based on past search trajectories. Besides, they effectively leverage the reasoning and decision-making capabilities of LLMs to improve the inner-task transfer propagability.

**Weaknesses:**

1. The use of LLM as an evolutionary operator (powered by prompt engineering) is interesting, similar ideas such as "Evolution through Large Models (ELM)" and [1-2] have been proposed. The paper shows a possible pipeline of integrating LLM into the co-design process of VSR, but does not provide a deeper analysis about "Why LLM works well?". The black-box nature of LLMs can make it challenging to understand the reasoning behind the generated designs, Adding more explanations in the LLM's decision-making process would be beneficial.

2. The paper mentions experimenting with different temperatures but does not provide a detailed sensitivity analysis of different prompts. In my opinion, the explanation of the intuition of your designed prompts is more important than the proposed pipeline.

3. This paper is missing a comparison with some important baseline algorithms.

4. The test tasks chosen for this paper are too simple to demonstrate the superiority and validity of large language models.

References

[1] Lange, Robert, Yingtao Tian, and Yujin Tang. "Large language models as evolution strategies." Proceedings of the Genetic and Evolutionary Computation Conference Companion. 2024.

[2]Hemberg, Erik, Stephen Moskal, and Una-May O’Reilly. "Evolving code with a large language model." Genetic Programming and Evolvable Machines 25.2 (2024): 21.

**Questions:**

1. The testing tasks such as Walker-v0 and Carrier-v0 used in the paper are too simple, can you test your method on more complex tasks ("Climber-v0", "Catcher-v0", "Thrower-v0" and "Lifter-v0"), which I think are more suitable to answer the question "Does LLM really have an advantage ?"

2. Can large language models really align a robot's design with its task performance? Is it enough to just use prompt engineering for more complex tasks? Can it be used as a surrogate model to predict the performance of a robot design? Can the authors give some explanations?

3. Can the current framework scale to more complex and larger robot designs (10x10 design space for walking)? If not, what are the potential bottlenecks? In larger design space (10 x 10), does LLM still work well? For some tasks, random combinations of voxels generated by LLM or evolutionary operators don't always work well.

4. To further improve this paper, it is better to show the designed robots by LLM and add analysis of the differences between llm-generated robot designs and  GA-generated robot designs.

5. While the paper demonstrates inter-task knowledge transfer, how well does LASER generalize to tasks that are significantly different from the ones used in the experiments? What are the limitations of this generalization?

6. The authors need to compare their approach with those that also use LLM as a mutation operator， such as openELM (Evolution through Large Models (ELM)) and more recent brain-body co-design methods (does not use LLM) which also use EvoGym platform, to show the effectiveness.

**Details Of Ethics Concerns:**

No.

---

> ### Author Response · Authors · 2024-11-26
> **Response to Reviewer D75R (1/4)**
>
> Dear reviewer,
>
> Thank you for your constructive feedback. The following is our detailed response to the questions you raised. Please let us know if you have any trouble accessing our revised paper.
>
> >**Weakness 1**: The paper does not provide a deeper analysis about “why LLM works well”. The black-box nature of LLMs can make it challenging to understand the reasoning behind the generated designs. Adding more explanations in the LLM’s decision-making process would be beneficial.
>
> >**Question 2**: Can LLMs really align a robot’s design with its task performance? Is it enough to just use prompt engineering for more complex tasks? Can it be used as a surrogate model to predict the performance of a robot design?
>
> **Response**:
> Thank you for your thoughtful feedback. In response to Weakness 1, we indeed included LLM reasoning in our preliminary experiments, where the LLM was prompted to explicitly explain its design choices (similar to chain-of-thought). However, we observed no significant performance gains from this practice. Consequently, we opted to remove the reasoning process to streamline evolution and reduce computational costs.
>
> However, we would like to clarify that **our approach is capable of affording higher interpretability**. To address your concerns, we have included additional results in Appendix J of revised paper, where the LLM is explicitly instructed to explain its decision-making process, rather than functioning as a black box. The explanations provided are insightful and reasonable, **revealing the advantageous structures present in high-performing designs**. This directly answers the question of whether LLMs can align robot designs with task performances, and sheds light on why LLMs work well. Specifically, LLMs are able to **identify favorable voxel assembly patterns** within designs and **leverage these insights to generate improved offspring solutions**. In response to Question 2, we have also tested our approach in more complex tasks and found that it continues to demonstrate notable advantages (as discussed in our answer to Question 1 and Appendix E).
>
> Regarding the use of LLMs as surrogate models, prior work has indeed explored this approach (Liu et al., 2024), where LLMs are employed to predict objective function values for computing the acquisition function in Bayesian Optimization. However, this study focus on a hyper-parameter fine-tuning task with less than ten decision variables. Another work in this line focuses on even simpler tasks, specifically two-dimensional mathematical functions (Hao et al., 2024). We speculate that LLMs may not perform as well in more complex optimization problems, particularly those involving high-dimensional, non-linear functional mappings, such as the voxel-based soft robot design in our study.
>
> Given these challenges, we argue that LLMs may not be suitable as direct function approximators for such complex problems. Instead, it would be more advisable to use LLMs as search operators, as this approach does not rely on precise function approximation, thereby avoiding the risk of mistakenly ruling out promising solutions. Meanwhile, it still leverages the reasoning capabilities of LLMs and the insights gleaned from successful solutions to expedite the evolutionary search process.

---

> ### Author Response · Authors · 2024-11-26
> **Response to Reviewer D75R (2/4)**
>
> >**Weakness 2**: The paper does not provide a detailed sensitivity analysis of different prompts. In my opinion, the explanation of the intuition of your designed prompts is more important that the proposed pipeline.
>
> **Response**:
> Thank you for raising this point. Our prompt design consists of three major components: task-related metadata, elite design-fitness pairs, and target fitness. The **task-related metadata** primarily includes descriptions of task objectives and the simulation environment. This component is largely derived from the official documents of EvoGym (Bhatia et al., 2021), with **minimal modifications**. This metadata, which is often overlooked in previous works on LLM-aided robot design, serves two main purposes: to ground the evolutionary process in the specific context of the problem, and to facilitate the transfer of knowledge between different tasks. The second component consists of **elite design-fitness pairs** previously evaluated, where the designs are sorted according to their fitness in ascending order. This sorting is intended to leverage the **pattern-completion capabilities** of LLMs, a technique shown to be effective in prior research (Lange et al., 2024; Yang et al., 2024). The third component, the **target fitness** (referred to as the “just-ask query” in Lim et al. (2024)), is introduced as a means of aligning the LLM’s outputs with our desired results. We would like to note that the phrasing of these components is **intentionally left simple and intuitive**, and no special techniques of prompt engineering were applied. As such, our experimental results possess a certain level of **robustness** and do not hinge on the specifics of prompt designs. However, it would be a promising direction to integrate various prompting techniques, such as chain-of-thought (Wei et al., 2022) and tree-of-thought (Yao et al., 2024), into our framework for better performances.
>
> To further highlight the significance of the individual components in our prompt and to complement the intuitive explanations provided, we conducted additional ablation studies. We found that removing each of the individual components led to performance drops. The just-ask query is proven the most essential, while simulation description and ordering play less important roles. The detailed results are reported in Appendix M of revised paper.
> _____
> >**Weakness 3**: The paper is missing a comparison with some important baseline algorithms.
> Question 6: The authors need to compare their approach with those that also use LLM as a mutation operator, and more recent brain-body co-design methods that do not use LLM but also use EvoGym platform.
>
> **Response**:
> We appreciate your feedback and have incorporated two additional baselines into our comparative analysis. The first is OPRO (Yang et al., 2024), a recent LLM-aided evolutionary strategy, which we adapted for VSR design. The second is MorphVAE (Song et al., 2024), a state-of-the-art co-design method that does not utilize LLM but also uses the EvoGym platform. As shown in Appendix G of revised paper, our approach **still outperforms these baselines**, further demonstrating its effectiveness.

---

> ### Author Response · Authors · 2024-11-26
> **Response to Reviewer D75R (3/4)**
>
> >**Weakness 4**: The test tasks chosen for this paper are too simple to demonstrate the superiority and validity of LLMs.
>
> >**Question 1**: The testing tasks such as Walker-v0 and Carrier-v0 used in the paper are too simple. Can you test your method on more complex tasks, which I think are more suitable to answer the question “Does LLM really have an advantage?”
>
> **Response**:
> Thank you for your valuable feedback. In response, we evaluated LASeR, along with LLM-Tuner (the most competitive baseline), on an additional task namely **Catcher-v0**, which is among the most challenging ones in the EvoGym task suite. The result, which can be found in Appendix E of paper, shows that our approach **continues to demonstrate notable advantages**, even in more complex task settings.
> _____
> >**Question 3**: Can the current framework scale to more complex and larger robot designs (10x10 design space for walking)? If not, what are the potential bottlenecks? In larger design spaces (10x10), does LLM still work well? For some tasks, random combinations of voxels generated by LLM or evolutionary operators do not always work well.
>
> **Response**:
> We appreciate your insightful comment. To evaluate the scalability of our approach to larger design spaces, we tested both LASeR and LLM-Tuner (the most competitive baseline) on 10x10 Walker-v0, with results provided in Appendix I of paper). Our findings demonstrate that LASeR **continues to outperform the baseline** in terms of optimization efficiency, even in this larger design space. We attribute this success to the unique capabilities of LLMs, which **do not rely on random mutations** (as seen in genetic algorithms and other heuristics). Instead, LLMs leverage their reasoning capabilities to identify favorable voxel assembly patterns (such as effective use of actuators) within high-performing designs. Based on these insights, they carry out more ***informed* mutations and recombinations** to generate offspring solutions (Please refer to Appendix J for LLM’s explanation of its decision process). It is worth noting that **the 10x10 configuration results in a design space that is** $2.65\times 10^{52}$ **times larger than the 5x5 case**, due to combinatorial explosion. Therefore, the promising results indicate a remarkable potential of our approach to scale to even larger and more complex robot design problems.
>
> We note that the current results (as reported in Appendix I) are averaged across two independent runs. We planned to conduct three runs but there remains one of them unfinished. We would post the complete results once they are available. For your reference, we notice that our experiments on 10x10 Walker-v0 take on average **1.5 times longer** than the 5x5 case.

---

> ### Author Response · Authors · 2024-11-26
> **Response to Reviewer D75R (4/4)**
>
> >**Question 4**: To further improve this paper, it is better to show the designed robots by LLM and add analysis of the differences between LLM-generated and GA-generated robot designs.
>
> **Response**:
> We greatly appreciate your suggestion. In response, we have included visualizations of the robot designs evolved by different algorithms in Appendix K of revised paper. Most notably, the robot designs obtained by LASeR exhibit **readily observable diversity**, which supports the quantitative results presented in Section 3.2.1.
> _____
> >**Question 5**: While the paper demonstrates inter-task knowledge transfer, how well does LASeR generalize to tasks that are significantly different from the ones used in the experiments? What are the limitations of this generalization?
>
> **Response**:
> Thank you for the thought-provoking question. In our paper we focused on transferring design experience between task instances that are intuitively similar. These experimental designs are largely based on the structural similarities between tasks as revealed in Wang et al. (2023).  Here we demonstrate that **the prior knowledge of task relationships is not strictly necessary for successful inter-task generalization**. Specifically, we conducted additional zero-shot experiments where LLM was given elite samples of Walker-v0, but instructed to propose designs for Jumper-v0, which is a significantly different task (Appendix O of revised paper). Remarkably, the LLM was able to dig deeper into the underlying inter-task associations and identify rather **general, low-level design principles** (such as the importance of actuators for shape change and rigid voxels for structure stability) that are still relevant to the new task. According to the results presented in Appendix O, **the zero-shot designs still outperform randomly sampled ones**. These results suggest that LLMs possess substantial potential for generalizing design experience across seemingly different optimization problems, as long as they share some common ground and are not completely irrelevant to one another.
> _____
> Thank you again for your time and effort spent reviewing this paper! Your suggestions have been really conducive to our revision. Please let us know if your questions and concerns have been fully addressed.
>
> **References**:
>
> [1]Bhatia, Jagdeep, et al. "Evolution gym: A large-scale benchmark for evolving soft robots." Advances in Neural Information Processing Systems 34 (2021): 2201-2214.
>
> [2] Hao, Hao, Xiaoqun Zhang, and Aimin Zhou. "Large Language Models as Surrogate Models in Evolutionary Algorithms: A Preliminary Study." arXiv preprint arXiv:2406.10675 (2024).
>
> [3] Lange, Robert, Yingtao Tian, and Yujin Tang. "Large language models as evolution strategies." Proceedings of the Genetic and Evolutionary Computation Conference Companion. 2024.
>
> [4] Lim, Bryan, Manon Flageat, and Antoine Cully. "Large Language Models as In-context AI Generators for Quality-Diversity." arXiv preprint arXiv:2404.15794 (2024).
>
> [5] Liu, Tennison, et al. "Large Language Models to Enhance Bayesian Optimization." The Twelfth International Conference on Learning Representations. 2024.
>
> [6] Song, Junru, et al. "MorphVAE: Advancing Morphological Design of Voxel-Based Soft Robots with Variational Autoencoders." Proceedings of the AAAI Conference on Artificial Intelligence. Vol. 38. No. 9. 2024.
>
> [7] Wang, Yuxing, et al. "PreCo: Enhancing Generalization in Co-Design of Modular Soft Robots via Brain-Body Pre-Training." Conference on Robot Learning. PMLR, 2023.
>
> [8] Wei, Jason, et al. "Chain-of-thought prompting elicits reasoning in large language models." Advances in neural information processing systems 35 (2022): 24824-24837.
>
> [9] Yang, Chengrun, et al. “Large Language Models as Optimizers.” The Twelfth International Conference on Learning Representations. 2024.
>
> [10] Yao, Shunyu, et al. "Tree of thoughts: Deliberate problem solving with large language models." Advances in Neural Information Processing Systems 36 (2024).

---

> ### Author Response · Authors · 2024-11-29
> **We look forward to your feedback!**
>
> Dear Reviewer D75R,
>
> Thank you so much for your thoughtful suggestions. We have made revisions in our re-uploaded paper according to your feedback, and the changes are outlined as follows. We believe these adjustments would more effectively clarify and highlight our contributions.
>
> - Included demonstrations of interpretable LLM decision-making processes in Appendix F, J and O;
>
> - Included finer-grained ablation studies on our prompt, as well as the intuitions behind the prompt design, in Appendix M;
>
> - Included comparisons with two more state-of-the-art baselines in Appendix G;
>
> - Included evaluations on a larger design space in Appendix J, as well as on a more complex task in Appendix E;
>
> - Visualized robot designs evolved by different algorithms in Appendix K to enable easier comparison;
> - Evaluated the capability of our approach to transfer experience across different task instances in Appendix O.
>
> We hope we have addressed all of your concerns regarding the value of our work. With the discussion period deadline approaching, we eagerly look forward to any further questions or comments you may have. We sincerely hope that our responses and revisions will assist you in re-evaluating our paper. Once again, we deeply appreciate the time and effort you have dedicated into reviewing our work. Happy Thanksgiving!

---

> > ### Comment · Reviewer_D75R · 2024-12-02
> > **Thanks for feedback**
> >
> > Happy thanksgiving, and thank you for the feedback. I really appreciate the efforts made by authors and decide to raise my score to 5. However, my concerns about this work have not been fully addressed.
> >
> > 1. In my opinion, the most suitable baseline for this paper is ELM, which also uses LLM-based mutation operators (not with the reflection mechanism), but the authors didn't use it.
> >
> > 2. The main purpose of introducing diversity must be clarified. Is the diversity introduced to improve the final co-design quality or sth? Sometimes, to achieve a high co-design performance, we do not need too much diversity.
> >
> > 3. The focus of this paper should be on using LLMs to solve robot design problems, so in the first part of the paper, I think a suitable logic would be to first describe the problems with robot design, what are the problems with current solutions, and how large models happen to be well suited to solving these problems due to their own characteristics, such as exploring in a wide range of design spaces, but also some limitations. In order to address these limitations, LASeR is proposed in this paper.

---

> > > ### Author Response · Authors · 2024-12-02
> > > **Further Response to Reviewer D75R (2/4)**
> > >
> > > **Response to 2 and 3:**
> > >
> > > Thank you very much for your suggestions. We agree that the paper will benefit from further clarification regarding the main purpose of introducing diversity. We also concur that a description of the robot design problem should precede the introduction of large language models to foster a more rational narrative logic. Following your advice, we have re-organized the first three paragraphs of the introduction. Since the deadline of paper uploading has passed, we include the revised version below.

---

> > > ### Author Response · Authors · 2024-12-02
> > > **Further Response to Reviewer D75R (4/4)**
> > >
> > > **We hope that the above responses have fully addressed your questions and concerns. Once again, please allow us to express our sincere gratitude for your time and effort dedicated into reviewing our paper.**
> > >
> > > **References:**
> > >
> > > [1] Achiam, Josh, et al. "Gpt-4 technical report." arXiv preprint arXiv:2303.08774 (2023).
> > >
> > > [2] Bhatia, Jagdeep,et al. "Evolution gym: A large-scale benchmark for evolving soft robots." Advances in Neural Information Processing Systems 34 (2021): 2201-2214.
> > >
> > > [3] Cheney, Nick, et al. "Unshackling evolution: evolving soft robots with multiple materials and a powerful generative encoding." ACM SIGEVOlution 7.1 (2014): 11-23.
> > >
> > > [4] Brahmachary, Shuvayan, et al. "Large Language Model-Based Evolutionary Optimizer: Reasoning with elitism." arXiv preprint arXiv:2403.02054 (2024).
> > >
> > > [5] Chocron, Olivier, and Philippe Bidaud. "Evolutionary algorithms in kinematic design of robotic systems." Proceedings of the 1997 IEEE/RSJ International Conference on Intelligent Robot and Systems,1997.
> > >
> > > [6] Tilman, David, et al. 2017. Future threats to biodiversity and pathways to their prevention. Nature 546, 7656 (2017), 73–81.
> > >
> > > [7] Hiller, Jonathan, and Hod Lipson. "Automatic design and manufacture of soft robots." IEEE Transactions on Robotics 28.2 (2011): 457-466.
> > >
> > > [8] Hu, Jiaheng, et al. "Modular robot design optimization with generative adversarial networks." 2022 International Conference on Robotics and Automation (ICRA). IEEE, 2022.
> > >
> > > [9] Hu, Jiaheng, Julian Whitman, and Howie Choset. "GLSO: grammar-guided latent space optimization for sample-efficient robot design automation." Conference on Robot Learning. PMLR, 2023.
> > >
> > > [10] Huang, Beichen, et al. "Exploring the True Potential: Evaluating the Black-box Optimization Capability of Large Language Models." arXiv preprint arXiv:2404.06290 (2024a).
> > >
> > > [11] Huang, Sen, et al. "When Large Language Model Meets Optimization." arXiv preprint arXiv:2405.10098 (2024b).
> > >
> > > [12] Karine Miras, Eliseo Ferrante, and AE Eiben. 2020. Environmental influences on evolvable robots. PloS one 15, 5 (2020).
> > >
> > > [13] Lange, Robert, Yingtao Tian, and Yujin Tang. "Large language models as evolution strategies." Proceedings of the Genetic and Evolutionary Computation Conference Companion. 2024.
> > >
> > > [14] Lehman, Joel, et al. "Evolution through large models." Handbook of Evolutionary Machine Learning. Singapore: Springer Nature Singapore, 2023. 331-366.
> > >
> > > [15] Liu, Shengcai, et al. "Large language models as evolutionary optimizers." 2024 IEEE Congress on Evolutionary Computation (CEC). IEEE, 2024a.
> > >
> > > [16] Liu, Tennison, et al. "Large Language Models to Enhance Bayesian Optimization." The Twelfth International Conference on Learning Representations, 2024b.
> > >
> > > [17] Medvet, Eric, et al. "Biodiversity in evolved voxel-based soft robots." Proceedings of the Genetic and Evolutionary Computation Conference. 2021.
> > >
> > > [18] Morris, Clint, Michael Jurado, and Jason Zutty. "Llm guided evolution-the automation of models advancing models." Proceedings of the Genetic and Evolutionary Computation Conference. 2024.
> > >
> > > [19] Qiu, Kevin, et al. "RoboMorph: Evolving Robot Morphology using Large Language Models." arXiv preprint arXiv:2407.08626 (2024).
> > >
> > > [20] Romera-Paredes, Bernardino, et al. "Mathematical discoveries from program search with large language models." Nature 625.7995 (2024): 468-475.
> > >
> > > [21] Sims, Karl. "Evolving virtual creatures." Seminal Graphics Papers: Pushing the Boundaries, Volume 2. 2023. 699-706.
> > >
> > > [22] Leger, Chris. Darwin2K: An evolutionary approach to automated design for robotics. Vol. 574. Springer Science & Business Media, 2012.
> > >
> > > [23] Song, Junru, et al. "MorphVAE: Advancing Morphological Design of Voxel-Based Soft Robots with Variational Autoencoders." Proceedings of the AAAI Conference on Artificial Intelligence. Vol. 38. No. 9. 2024a.
> > >
> > > [24] Team, Gemini, et al. "Gemini: a family of highly capable multimodal models." arXiv preprint arXiv:2312.11805 (2023).
> > >
> > > [25] Team, InternLM. "Internlm: A multilingual language model with progressively enhanced capabilities." 2023-01-06)[2023-09-27]. https://github. com/InternLM/InternLM (2023).
> > >
> > > [26] Touvron, Hugo, et al. "Llama 2: Open foundation and fine-tuned chat models." arXiv preprint arXiv:2307.09288 (2023).
> > >
> > > [27] Tran, Thanh VT, and Truong Son Hy. "Protein design by directed evolution guided by large language models." IEEE Transactions on Evolutionary Computation (2024).
> > >
> > > [28] Wang, Tingwu, et al. "Neural graph evolution: Towards efficient automatic robot design." arXiv preprint arXiv:1906.05370 (2019).
> > >
> > > [29]Yang, Chengrun, et al. "Large Language Models as Optimizers." The Twelfth International Conference on Learning Representations, 2024.
> > >
> > > [30] Ye, Haoran, et al. "Large language models as hyper-heuristics for combinatorial optimization." arXiv preprint arXiv:2402.01145 (2024).
> > >
> > > [31] Zhang, Lechen. "CUDA-Accelerated Soft Robot Neural Evolution with Large Language Model Supervision." arXiv preprint arXiv:2405.00698 (2024).

---

> ### Author Response · Authors · 2024-12-02
> **Further Response to Reviewer D75R (1/4)**
>
> Dear Reviewer D75R,
>
> We greatly appreciate your insightful feedback, which is very helpful to our revision. Below is our response to the concerns you raised.
>
> **Response to 1:**
>
> We share your view that *Evolution Through Large Models* (ELM) is one of the most representative and to our knowledge, the earliest work on LLM-based evolutionary optimization. However, we did not include ELM as one of our baselines because it was originally proposed for designing Sodaracers through **code optimization**, which necessitates a program interface that maps between robot designs and Python codes. Such a mapping becomes **ambiguous and difficult to define** in the context of voxel-based soft robots (VSRs). Consequently, to further verify the effectiveness of our proposed method, we performed additional comparisons with another evolutionary approach that uses LLMs as mutation operators, **OPRO** (Yang et al., 2024), which is more recent and operates within the original solution space as we do. The results (presented in Appendix G in our re-uploaded paper) still demonstrate a clear advantage of LASeR. Nevertheless, to acknowledge the significance of this pioneering work, we have cited ELM (Lehman et al., 2023) in Section 2.1 and it will be one of our focuses in future work to devise an appropriate code-level solution space for VSRs, both to examine the potential impact of different solution representations and to allow for a more direct comparison with ELM.

---

> ### Author Response · Authors · 2024-12-02
> **Further Response to Reviewer D75R (3/4)**
>
> **Revised introduction: (the first three paragraphs)**
>
> *Robot design automation represents an established and persistent challenge in modern robotics, aiming to autonomously evolve robot morphology with minimal human intervention (Hu et al., 2022; 2023; Song et al., 2024a). Existing approaches predominantly rely on traditional evolutionary algorithms and are primarily focused on rigid robots (Chocron & Bidaud, 1997; Leger, 2012; Wang et al., 2019; Sims, 2023; Hu et al., 2023). Recently, modular soft robots have attracted considerable attention due to their remarkable versatility, expressiveness, and biomimetic properties (Hiller & Lipson, 2011; Bhatia et al., 2021; Medvet et al., 2021). However, these advantages also introduce significant challenges to robot design automation. Specifically, modular soft robots often involve combinatorially vast design spaces and intricate interaction dynamics, rendering existing approaches prone to local optima. This calls for more efficient search algorithms that can navigate the vast design space while ensuring progressive improvement in functionality (Cheney et al., 2014; Song et al., 2024). Furthermore, most of current approaches rely heavily on problem-specific mathematical formulations and manually designed search heuristics, leaving under-explored the potential for a more accessible design process driven by natural language instructions.*
>
> *In recent years, large language models (LLMs) have demonstrated impressive reasoning, decision-making, and generalization capabilities (Achiam et al., 2023; Touvron et al., 2023; Team et al., 2023; Team, 2023), sparking a flurry of research interest in their application to optimization problems. Earlier efforts embarked on employing LLMs to aid traditional search heuristics, such as selecting parent solutions for mutation and crossover (Liu et al., 2024a; Ye et al., 2024), or serving as surrogate models and candidate samplers in Bayesian Optimization (Liu et al., 2024b). More recent studies have explored the use of LLMs as “intelligent search operators”. By receiving previously found solutions through prompts, LLMs effectively leverage their in-context learning and pattern-completion capabilities to iteratively propose improved candidate solutions (Brahmachary et al., 2024; Huang et al., 2024b; Yang et al., 2024; Morris et al., 2024; Romera-Paredes et al., 2024; Lange et al., 2024). These LLM-aided evolutionary frameworks have also shown great promise in reducing reliance on handcrafted search heuristics, facilitating convenient problem specification in natural language and rendering evolutionary processes more interpretable. To date, they have showcased proficiency in both classic optimization problems (Liu et al., 2024a; Brahmachary et al., 2024; Huang et al., 2024a) and real-world applications (Morris et al., 2024; Romera-Paredes et al., 2024; Lange et al., 2024; Tran & Hy, 2024).*
>
> *With the promise of higher efficiency and interpretability for evolutionary computation, LLMs have also made their way into the realm of robot design automation. To our best knowledge, the only relevant studies are Zhang (2024), Qiu et al. (2024) and Lehman et al. (2023). While Zhang (2024) explores the use of LLMs to tune hyperparameters of traditional evolutionary algorithms, the latter two pioneer the use of LLMs as search operators for robot design. Nonetheless, these studies still bear some major limitations. Firstly, as highlighted by several related works, LLMs often struggle to balance exploration and exploitation, leading to inferior solution diversity (Huang et al., 2024b; Tran & Hy, 2024). This issue remains largely unaddressed in the aforementioned studies. **Enhancing diversity in evolved solutions is especially relevant in robot design automation as it is critical for adapting robot ecosystems to hazardous and dynamic environments (David et al., 2017; Medvet et al., 2021). Achieving both diversity and quality has long been a dilemma in evolutionary robotics (Karine et al., 2020; Medvet et al., 2021), and it remains to be investigated whether the reasoning capabilities of LLMs could be exploited to promote more intelligent exploratory behaviors in the design space.** Secondly, current LLM-aided evolutionary approaches generally lack strong connections to the specific context of real-world problems, resulting in suboptimal performances and solutions that can not generalize well. As it is common to have access to task-related metadata and a repository of pre-designed robots when designing for new applications, it is highly pertinent to ground evolutionary search in such contextual information, so as to enable inter-task experience transfer and foster more generalizable design processes.*

---

### Official Review · Reviewer_pDhF · 2024-11-04

**Soundness:** 3
**Presentation:** 3
**Contribution:** 2
**Rating:** 6
**Confidence:** 3

**Summary:**

The paper presents LASeR, an framework leveraging Large Language Models (LLMs) to optimize robot design with evolutionary algorithms. The proposed approach addresses the limitations of existing LLM-driven optimization techniques, such as limited solution diversity and generalizability across tasks. LLM is employed as the intelligent search operator and diversity reflecter, instead of a tool for hyperparameter tuning. By introducing a Diversity Reflection Mechanism (DiRect), LASeR refines the exploration-exploitation tradeoff, enhancing diversity and performance of the robot design automation tasks, comparted to baselines. Through task-grounded prompts, LASeR also enables effective knowledge transfer across different robot design tasks.

**Strengths:**

* The paper is clear and well-structured.
* The idea of applying LLMs in generating offspring for robot design evolutionary algorithms is interesting.
* Extensive experimental results on EvoGym are provided to validate that the proposed method outforms baselines in both design efficiency/performance and diversity.

**Weaknesses:**

* The experiments of the paper did not mention the time taken of the LLM-based methods for the evolutionary algorithm, which should be considered into the evaluation of robot design efficiency.
* The similarity threshold seems to be an important hyperparameter for the proposed method, while it is not discussed in the paper.
* The experiments in the paper are restricted to relatively simple voxel-based soft robots within predefined settings.
* The core method does not involve learning or fine-tuning for LLMs besides PPO utilized for the fitness evaluation.

**Questions:**

* What considerations were taken when selecting similarity thresholds? How do you balance the diversity and performance of the generated designs?
* In Section 3.2.1, you mentioned that the fitness performance of robot designs would not change significantly after being modified by DiRect. Do you have quantative results to support this conclusion?
* How does the computation time of LASeR compared to the baselines, including LLM-Tuner and the ones without using LLMs?

---

> ### Author Response · Authors · 2024-11-26
> **Response to Reviewer pDhF (1/2)**
>
> Dear reviewer,
>
> Thank you for your constructive feedback. Below is our detailed response to the questions you raised. Please let us know if you have any trouble accessing our revised paper.
>
> >**Weakness 1**: The experiments of the paper did not mention the time taken of the LLM-based methods for the evolutionary algorithm, which should be considered into the evaluation of robot design efficiency.
>
> >**Question 3**: How does the computation time of LASeR compare to the baselines, including LLM-Tuner and the ones without using LLMs?
>
> **Response:**
> Thank you for pointing out this important issue. According to the data released on LLM Leaderboard (https://artificialanalysis.ai/leaderboards/models), for GPT-4o-mini, the median rate of output token generation is 99.8 tokens per second, and the latency (i.e. time to first token) is reported as 0.5 seconds. Given that LASeR makes an average of 130 API calls per generation, with each call involving approximately 180 output tokens (here we assume the worst case where each newly generated robot design triggers DiRect), this results in an overhead of around 5 minutes per generation, or 5 hours in total.
>
> The latency issue could be mitigated with locally deployed LLMs, which are less affected by network delays and request queuing. However, we believe it is more pertinent to compare the overall running time of different methods, with optimization efficiency taken into account. Specifically, by checking the log messages of our programs, we find that, for Carrier-v0, in order to reach the same level of fitness, LASeR requires **7 hours**, in contrast to the most competitive baseline,  LLM-Tuner, which takes **15 hours**. For Pusher-v0, the difference is greater: LASeR requires **11 hours**, whereas LLM-Tuner takes **46 hours**. On Walker-v0, LASeR is even capable of **reaching a fitness unattainable by baselines**. Hence, the rapid convergence enabled by LLMs outweighs the additional computation overhead, **rendering the latter perfectly worthwhile**. We have also included this analysis of computational efficiency in Appendix Q of revised paper.
> _____
> >**Weakness 2**: The similarity threshold seems to be an important hyperparameter for the proposed method, while it is not discussed in the paper.
>
> >**Question 1**: What considerations were taken when selecting similarity thresholds? How do you balance the diversity and performance of the generated designs?
>
> **Response:**
> The similarity threshold is indeed a crucial hyper-parameter that modulates the performance of LASeR. The choice of this threshold reflects the extent of diversity that one expects to see in the evolved solutions, and therefore **should be driven by the user’s specific preferences**. For instance, setting the threshold to 20 (as we did in our experiments) means that if a newly generated robot design shares more than 20 identical voxels with any existing solutions, it will undergo modifications by DiRect.
>
> There are some **general principles** for choosing this parameter. These principles are supported by our additional experiments with several different values of threshold (as shown in Appendix N of revised paper). High similarity thresholds, like threshold=23, are generally not recommended, as they would hinder the beneficial exploration enabled by LLMs. Conversely, excessively low thresholds (such as threshold=15) might increase diversity but also risk overly aggressive exploration that compromises functionality and, in turn, harms optimization efficiency. We believe this is due to the poor extrapolation performance of LLMs when required to propose robot designs that are much different from given examples. Any moderate values in the middle should give rise to desirable performances. In fact, our findings suggest that a threshold of 18 leads to further performance gains beyond 20, which we have chosen in our study. However, we note that lower thresholds also more frequently trigger DiRect, which means more LLM API calls. Hence, **the threshold choice also involves a trade-off between evolutionary performance** (including both optimization efficiency and diversity) and **computational costs**, and **should be considered case by case**. We believe adaptive threshold scheduling, based on problem specifics and evolutionary outcomes, could be a promising direction for future research.
> _____

---

> ### Author Response · Authors · 2024-11-26
> **Response to Reviewer pDhF (2/2)**
>
> >**Weakness 3**: The experiments in the paper are restricted to relatively simple voxel-based soft robots within predefined settings.
>
> **Response:**
> In this paper, we adopt the commonly used setup from previous research on voxel-based soft robots (VSRs) (Song et al., 2024; Saito and Oka, 2024; Dong et al., 2024; Wang et al., 2023; Bhatia et al., 2021), specifically the 5x5 design space with 5 distinct materials. This configuration yields a combinatorially vast search space, encompassing approximately $2.98\times10^{17})$ possible designs. This design space has proven **sufficiently expressive, allowing for the emergence of complex and diverse morphological structures**. We adhere to this setup also to facilitate a **direct comparison** with the majority of studies on VSR. That being said, our approach is fully scalable to larger design spaces and could also applied to other types of robots. We conducted an additional experiment in which we used LASeR to evolve 10x10 Walkers (Appendix I of revised paper), and **the advantage of our approach remains evident**. We note that the current results are averaged across two independent runs. We planned to conduct three runs but there remains one of them unfinished. We would post the complete results once they are available. For your reference, we notice that our experiments on 10x10 Walker-v0 take on average **1.5 times longer** than the 5x5 case. Evaluating the potential of LLMs for larger-scale and more intricate robot design problems will be one of our focuses in future work.
> _____
> >**Weakness 4**: The core method does not involve learning or fine-tuning for LLMs besides PPO utilized for the fitness evaluation.
>
> **Response:**
> In this work, we opt for **the most cost-effective** method of leveraging LLMs for robot design automation, which is through in-context learning or instruction tuning. Our experimental results demonstrate that pre-trained LLMs, once appropriately prompted, can **already achieve excellent optimization performance**. Fine-tuning the LLM parameters would incur substantially higher costs, both in terms of computational resources and the need for a large, carefully curated dataset. Moreover, fine-tuning introduces the risk of issues such as overfitting and catastrophic forgetting, which might negatively impact performances. Nevertheless, we greatly appreciate your insightful comment. We recognize that this is a promising avenue for future exploration, and fine-tuning general-purpose LLMs for various combinatorial optimization tasks represents a valuable research direction. We have included a discussion of this prospect in Appendix P of our revised paper.
> _____
>
> >**Question 2**: In Section 3.2.1, you mentioned that the fitness performance of robot designs would not change significantly after being modified by DiRect. Do you have quantitative results to support this conclusion?
>
> **Response**:
> Thank you for your value feedback. We have now included the quantitative results that led to our argument in Section 3.2.1. Concretely, we took Walker-v0 as an example and conducted a two-tailed Student’s $t$-test to compare the fitness of robot designs before and after DiRect modification. The resulting $p$-value was **0.19**, which exceeds the 0.05 significance threshold, indicating no significant change in fitness. Additionally, we replaced the DiRect mechanism with random mutations, in which case the $p$-value was **less than 0.001**, demonstrating a significant decrease in fitness compared to our diversity reflection mechanism. For further discussion and experimental results of LLMs serving as ***intelligent* mutation operators**, the reviewer is referred to Appendix F of revised paper.
> _____
> Thank you again for your time and effort spent reviewing this paper! Your suggestions have been really conducive to our revision. Please let us know if your questions and concerns have been fully addressed.
>
> **References**:
>
> [1] Bhatia, Jagdeep, et al. "Evolution gym: A large-scale benchmark for evolving soft robots." Advances in Neural Information Processing Systems 34 (2021): 2201-2214.
>
> [2] Dong, Heng, Junyu Zhang, and Chongjie Zhang. "Leveraging Hyperbolic Embeddings for Coarse-to-Fine Robot Design." The Twelfth International Conference on Learning Representations. 2024.
>
> [3] Saito, Takumi, and Mizuki Oka. "Effective Design and Interpretation in Voxel-Based Soft Robotics: A Part Assembly Approach with Bayesian Optimization." Artificial Life Conference Proceedings 36. Vol. 2024. No. 1. One Rogers Street, Cambridge, MA 02142-1209, USA journals-info@ mit. edu: MIT Press, 2024.
>
> [4] Song, Junru, et al. "MorphVAE: Advancing Morphological Design of Voxel-Based Soft Robots with Variational Autoencoders." Proceedings of the AAAI Conference on Artificial Intelligence. Vol. 38. No. 9. 2024.
>
> [5] Wang, Yuxing, et al. "PreCo: Enhancing Generalization in Co-Design of Modular Soft Robots via Brain-Body Pre-Training." Conference on Robot Learning. PMLR, 2023.

---

> ### Author Response · Authors · 2024-11-29
> **We look forward to your feedback!**
>
> Dear Reviewer pDhF,
>
> We very much appreciate your thoughtful suggestions and have made modifications accordingly in our re-uploaded paper. The updates that we have made in response to your feedback are listed as follows, which we believe would better illuminate our contributions.
>
> - Included an analysis of computational efficiency in Appendix Q;
>
> - Included a further discussion on the principles of choosing the similarity threshold, together with additional experimental results with varying thresholds, in Appendix N;
>
> - Added experimental results on Walker-v0 with a 10x10 body size in Appendix I, in order to demonstrate the scalability of our approach to larger design spaces;
>
> - Explained the prospect of fine-tuning LLMs for general-purpose combinatorial optimization as a promising future direction in Appendix P;
>
> - Included quantitative results to prove the role of LLMs as intelligent mutation operators in Appendix F.
>
> We hope that we have fully addressed all your concerns regarding the value of our work. With the deadline of the discussion period approaching, we eagerly look forward to any additional questions or comments you may have. We genuinely hope that our responses and revisions could help you re-evaluate our paper. Once again, we greatly appreciate your time and effort dedicated to reviewing our paper. Happy Thanksgiving!

---

> > ### Comment · Reviewer_pDhF · 2024-12-02
> >
> > I sincerely thank the authors for the detailed responses and additional experimental results, which solve my main concerns. I have raised my score accordingly.

---

> > > ### Author Response · Authors · 2024-12-04
> > >
> > > We greatly appreciate your positive feedback. Thank you again for your time and efforts!

---

### Author Response · Authors · 2024-11-26
**General Response**

Dear reviewers, area chairs and program chairs,

We sincerely thank you for the time and effort that you spent reviewing our paper. We very much appreciate the reviewers’ recognition of our contributions, which we briefly summarize as follows.

>**Novelty:** “The proposed framework is novel, and the usage of LLMs in robot design and their evolution is under-researched” (Reviewer 5ABx); “This paper demonstrates originality by developing a mechanism aimed at solving lack of diversity in LLM-aided robot design processes... The idea of inter-task reasoning also adds to originality” (Reviewer drYB).

>**Presentation:** “The paper is clear and well-structured” (Reviewer pDhF); “...clearly set out intended contributions, scientific hypotheses and experiments... Figures are high quality and aesthetically pleasant...covering a fairly broad portion of related literature” (Reviewer drYB); “The paper is well written and easy to follow” (Reviewer drYB).

>**Effectiveness and significance:** “...effectively leverage the reasoning and decision-making capabilities of LLMs to improve the inter-task transfer propagability” (Reviewer D75R). “The conclusions the paper makes, and its applications are relevant to the robot learning community” (Reviewer 5ABx); “Joint design and control of robotic systems is an important problem... research and development in this area is crucial to advance the field” (Reviewer drYB).

>**Experiments:** “Extensive experimental results are provided” (Reviewer pDhF); “The performed ablation studies are very interesting and insightful” (Reviewer 5ABx).

We refer the reviewers to the **individual responses** where we addressed your questions in detail. We have also made revisions to our paper accordingly (in blue font). Please let us know if your concerns are fully addressed or if there is any difficulty accessing our revised paper. Once again, please allow us to express our sincere gratitude for your valuable suggestions and feedback, which have been really conducive to our revisions.

---

### Meta-Review · Area_Chair_Wn2J · 2024-12-25

**Metareview:**

The paper introduces LASeR (Large Language Model-Aided Evolutionary Search for Robot Design Automation), which advances robot design optimization by using a novel reflection mechanism (DiRect) to balance exploration and exploitation, improve solution diversity, and enable inter-task reasoning for generalizable and zero-shot robot proposals. The method demonstrates superior performance in voxel-based soft robot experiments.

The reviewers acknowledged the paper's significant contributions, highlighting (1) the relevance and importance of the addressed problem, (2) the novelty and interest of the proposed idea, (3) extensive experimental validation, and (4) the clarity and structure of the presentation.

During the Author-Reviewer Discussion phase, the authors provided thorough and well-reasoned responses, successfully addressing most concerns and convincing some reviewers to raise their scores. The AC encourages the authors to carefully consider both pre- and post-rebuttal comments to address any remaining concerns in a future revision.

**Additional Comments On Reviewer Discussion:**

During the Reviewer Discussion phase, Reviewer D75R remained negative but did not engage to provide compelling arguments against the paper. After thoroughly reviewing the reviews, rebuttal, and discussion, the AC concludes that the authors have adequately addressed Reviewer D75R’s concerns and provided reasonable justifications. Therefore, the AC recommends accepting the paper.

---

### Decision · Program_Chairs · 2025-01-22

Accept (Poster)